# Large-scale assembly of isotropic nanofiber aerogels based on columnar-equiaxed crystal transition

Lei Li [1,2,9], Yiqian Zhou[1,9], Yang Gao[3,9], Xuning Feng[4], Fangshu Zhang[4], Weiwei Li [5] ✉, Bin Zhu [6], Ze Tian[1], Peixun Fan [1], Minlin Zhong [1], Huichang Niu[7], Shanyu Zhao [8], Xiaoding Wei [3] ✉, Jia Zhu [6] ✉ & Hui Wu [1] ✉

Ice-templating technology holds great potential to construct industrial porous materials from nanometers to the macroscopic scale for tailoring thermal, electronic, or acoustic transport. Herein, we describe a general ice-templating technology through freezing the material on a rotating cryogenic drum surface, crushing it, and then re-casting the nanofiber slurry. Through decoupling the ice nucleation and growth processes, we achieved the columnar-equiaxed crystal transition in the freezing procedure. The highly random stacking and integrating of equiaxed ice crystals can organize nanofibers into thousands of repeating microscale units with a tortuous channel topology. Owing to the spatially well-defined isotropic structure, the obtained $Al_2O_3 \cdot SiO_2$ nanofiber aerogels exhibit ultralow thermal conductivity, superelasticity, good damage tolerance, and fatigue resistance. These features, together with their natural stability up to 1200 °C, make them highly robust for thermal insulation under extreme thermomechanical environments. Cascading thermal runaway propagation in a high-capacity lithium-ion battery module consisting of $LiNi_{0.8}Co_{0.1}Mn_{0.1}O_2$ cathode, with ultrahigh thermal shock power of 215 kW, can be completely prevented by a thin nanofiber aerogel layer. These findings not only establish a general production route for nanomaterial assemblies that is conventionally challenging, but also demonstrate a high-energy-density battery module configuration with a high safety standard that is critical for practical applications.

In the vast area of low-dimensional materials, pursuing synthesis and assembly system development to enable commercial products has been a long-term goal[1,2]. For instance, fluidized bed chemical vapor deposition (FB-CVD)[3], industrial solution blow spinning (SBS)[4,5], and needle-free electrospinning (NF-ES)[6] are some examples of high-throughput synthesis technologies for one-dimensional (1D) materials. Meanwhile, various methods have been applied to process nanofibers to bulk materials, including layer-by-layer stacking[7,8], solvothermal assembling[9], ice-templating[10], vacuum filtration[11], and 3D printing[12]. Specifically, ice-templating holds considerable promise balancing easy scalability, precise architectural control, and high versatility[13–16]. On the

shaping of macro-morphology, ice-templating can duplicate the original appearance of molds and regulate the arrangements, rendering open cellular[17,18] or lamellar networks[19]. In a typical unidirectional freeze-casting process, the ice crystals heterogeneously nucleate on a cryogenic surface and spring up in a preferred direction along the temperature gradient. However, templating from the generated columnar crystals, the prepared bulk material presents a structure anisotropy[20]. The hollow channels would become a highway for thermal, electronic, or acoustic transport[21,22].

Taking ceramic nanofiber assembly as an example, with reliable mechanical properties at ultralow/ultrahigh temperatures, they show

great potential in extreme thermal insulation[23–25]. Past attempts to modulate thermal conductivities focused on developing a family of aerogels based on designs of flexible building blocks with ultralight features. The heat conduction across hollow channels (referred to as radial heat transfer) could be effectively reduced with an low thermal conductivity of 0.02 W m$^{-1}$ K$^{-1}$ owing to the highly tortuous heat transfer path and inhibited air diffusion[26]. However, because the channel dimensions are much greater than the standard mean free path of air (69 nm), it fails to restrict the free movement of molecules and mitigate the gaseous heat conduction. Meanwhile, the aligned channel wall is also beneficial to solid heat conduction. As a result, the heat conduction along the hollow channels (referred to as axial heat transfer) is usually two times higher than the radial thermal conductivity[27]. A thermal conductivity of around 0.04 W m$^{-1}$ K$^{-1}$ could be achieved, equivalent to those of currently used insulators such as expanded polystyrene, polyurethane, fiberglass, and mineral wool[20]. Thus, more sophisticated designs are necessary to attain improved thermal insulation performance. In this regard, many strategies have been introduced to regulate micro-architecture designs in the freeze-casting process, such as wettability control[13], temperature gradient direction[28,29], additive modification[30,31], magnetic field assistance[32], and interface design[33,34].

However, in the above-mentioned freeze-casting processes, the slurry still starts freezing under single or double temperature gradients. These surface growth processes would gradually slow down or even stop, owing to the temperature gradient vanishing and the heat transfer reduction. The limited nucleation and growth process severely hinders the preparation efficiency and thickness scale of porous structures. Alternatively, dynamic freeze casting and ice particulate templating can achieve multi-point ice nucleation and growth[35–37]. The obtained uniform porous materials have been proven to be applicable to tissue scaffolds. These methods partly overcome the issues of low nucleation sites and growth efficiency in the freeze-casting process. However, such behavior, combined with process intensification and structural evolution, has not been verified nor extended to different material systems. Most critically of all, the fundamental question of how ice crystal transformation can allow for such a significant alteration in structure remains unanswered.

Here, a general, controllable crushed ice templating strategy has been developed to large-scale fabricate a kind of isotropic aerogels composed of spatially well-defined nanofiber assemblies (e.g., aluminosilicate nanofibers (ASNFs), polyacrylonitrile nanofibers (PNFs), p-aramid nanofibers (ANFs), cellulose nanofibers (CNFs), and graphite nanotubes (GNTs)). As a typical demonstration, in addition to the high-temperature tolerance, the ASNF aerogels deliver an impressively mechanical performance with superelasticity, negative Poisson's ratio, good fatigue resistance, arising from the entanglement and bonding of nanofiber network via the formation of amorphous AlBSi glass ceramics. Additionally, remarkable structure isotropy in optics and thermal conductivity is also observed. With the combined merits of lightweight, mechanical flexibility, and separate cellular structure, the aerogels exhibit high thermal insulating performances in extreme thermomechanical environments. The devastating thermal runaway propagation process in a battery module with a high-energy-density of 267.3 Wh kg$^{-1}$, including complicated mechanical and thermal shock behaviors, can be blocked by a 5-mm-thick ceramic nanofiber aerogel layer with negligible weight burden on the whole system, promising its great potentials in various thermal protection scenarios.

## Results

### Fabrication of isotropic nanofiber aerogels

The procedure for preparing isotropic nanofiber aerogels is schematically shown in Fig. 1a, b. The synthesis starts with the mass production of flexible nanofibers. For example, the ASNFs and PNFs can be fabricated by a multi-needle SBS method (Supplementary Fig. 1)[4]. The nanofiber was then dispersed in a suitable solution, rapidly frozen on a rotating cryogenic drum (RCD) surface at −20 °C and scraped into crushed ice (Supplementary Fig. 2 and Movie 1, see Supplementary Materials for details). The rotating speed was set as 50 rad min$^{-1}$, because a higher rotating speed would cause the incomplete freezing of the nanofiber dispersion. The resulting crushed ice exhibits irregular shapes with a diameter between 235 and 355 μm. To obtain bulk nanofiber aerogels, we introduced a re-casting step based on the stacking and interface-integrating of crushed ice crystals using the initial slurry. During this process, the crushed ice crystals would be further refined under the action of external force to ensure uniform distribution. The mixture was then transformed into a cryogenic environment and re-casted as an ice block. The ice block was freeze-dried and calcined at 600 °C to obtain nanofibrous aerogels. We showed that the nanofiber aerogel formation process was easily scaled in a linear fashion to over 1.2 m$^2$ area.

Moreover, the crushed ice casting (CIC) method can also construct lightweight structures from various low-dimensional materials. Owing to the rapid preparation and large proportion introduction of crushed ice, this method is less prone to particle sedimentation in preparing large-sized samples, exhibiting high compatibility with different low-dimensional materials compared to traditional methods. We successfully prepared 3D porous aerogels from the crushed ice containing ASNFs, PNFs, ANFs, CNFs, and GNTs (Fig. 1c, d). Figure 1e shows the stepwise enlarged view of ASNF aerogels. An interconnected separate cellular structure with bonded nanofiber networks and silica aerogels can be readily observed, which would endow the aerogels with robust mechanical and thermal performance. The structure with different compositions, morphologies, and temperature resistances has applications in emerging fields owing to varying combinations of mechanical, thermal, or other properties (Supplementary Fig. 3, see Supplementary Materials for details). Additionally, the ASNF aerogels can serve as a potential skeleton for different functional materials, such as carbon dots (CDs) and MXene, which effectively convert photoluminescence and electromagnetic shielding functions into macrostructures (Supplementary Fig. 4, see Supplementary Materials for details).

We further investigated the effect of density regulation and size enlargement on the mechanical properties of the resultant materials. The densities of the ASNF aerogels can be readily regulated by changing the concentration of the precursor slurry. The minimum density achieved was 0.59 mg cm$^{-3}$ (Supplementary Fig. 5). However, when the fiber concentration was too low (less than 0.1%), the nanofibers were challenging to connect with each other, and the obtained aerogels would be loose without detectable mechanical strength. When the nanofiber concentration was too high (higher than 1.8%), the nanofibers would be agglomerated and difficult to disperse evenly in the solution, which was incompatible with the CIC process. Moreover, the thermal conductivity and elastic modulus of the obtained nanofiber aerogel increased as the density increased (Supplementary Table 1). Eventually, to achieve both low thermal conductivity and high elastic modulus, a density of 5 mg cm$^{-3}$ was selected for test samples.

Supplementary Fig. 6a, b showed the snapshots and plots of compressive stress *versus* strain and the corresponding optical image series during the first compression cycle of ASNF aerogels. The aerogels can completely recover to the original state after withstanding a compressive strain ranging from 30 to 90%. Moreover, during the compression and release processes, our aerogels exhibit consistent negative Poisson's ratios (Supplementary Fig. 6c and Movie 2). The superelastic characteristics and negative Poisson's ratios were mainly derived from the bonded nanofiber networks and separate cellular structures. This may attract considerable attention for diverse applications, particularly in dynamic impact environments such as aerospace. Plots of the relative compressive moduli ($E/E_s$) versus the relative densities ($\rho/\rho_s$) for different ASNF aerogels showed the power law of $E/E_s \sim (\rho/\rho_s)^{2.3}$ (Supplementary Fig. 6d). The exponent found here

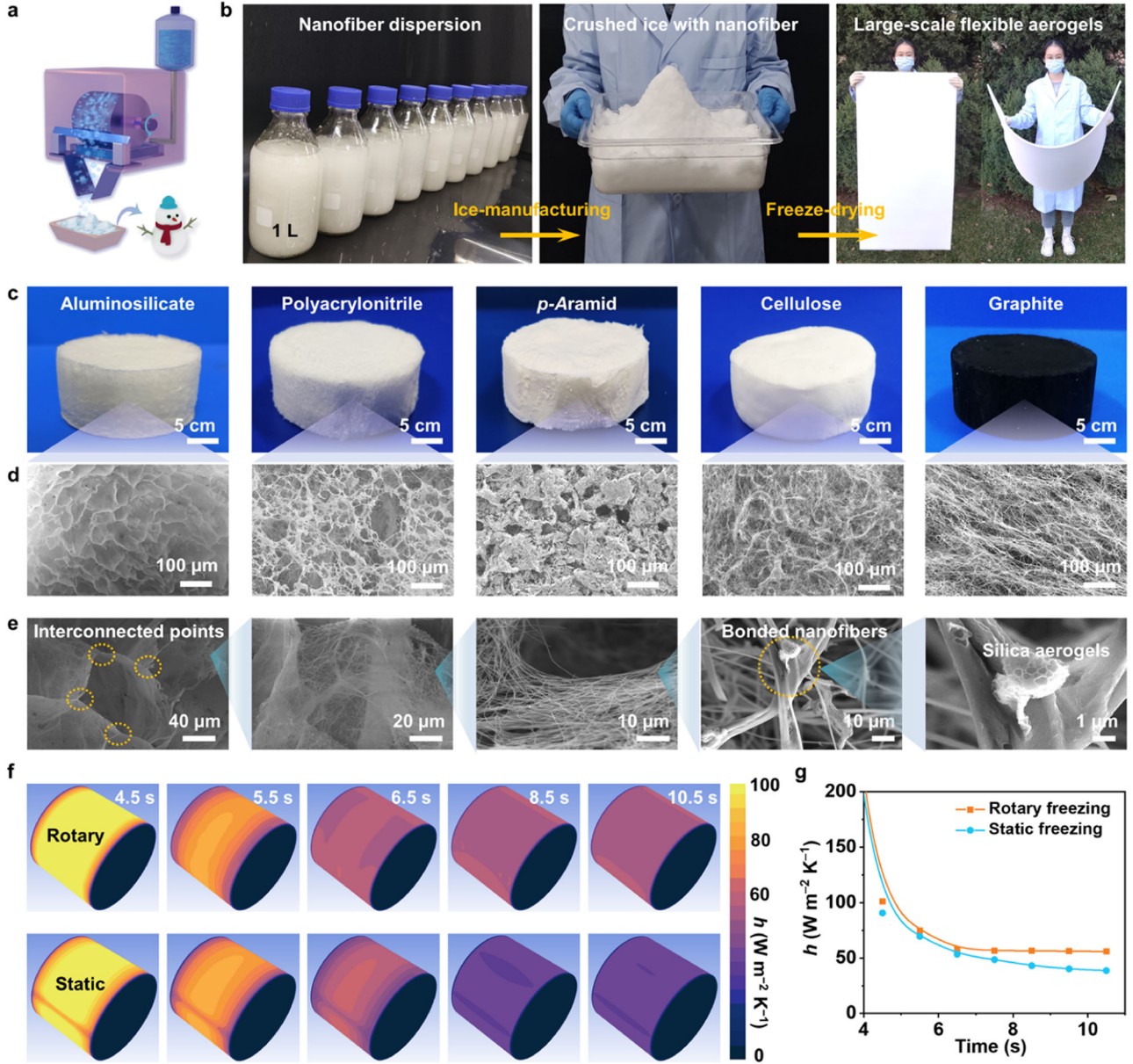

**Fig. 1 | Preparation and characterization of the nanofiber aerogels. a** Schematic illustration of the fabrication process of nanofiber aerogels using crushed ice casting (CIC) method. **b** Digital photograph of nanofiber dispersion, crushed ice, and structure-intact nanofiber aerogels of about 1.2 m² area. **c, d** Extension of a CIC method for various nanofiber aerogels from aluminosilicate nanofibers (ASNFs), polyacrylonitrile nanofibers (PNFs), *p*-aramid nanofibers (ANFs), cellulose nanofibers (CNFs), and graphite nanotubes (GNTs). **e** Morphology of ASNF aerogels at different magnifications, demonstrating the hierarchical nanofiber cellular architecture. **f** Computational fluid dynamics simulations on the surface of the rotary or static cryogenic drum. **g** Heat transfer coefficient comparison on the rotating drum surface on static and rotary state.

was close to those of most reported flexible structures, including mullite sponges[4], $Al_2O_3$ micro-lattices[38], CNT foams[39], Ni-P micro-lattices[40], and hBN aerogels[41], suggesting a bending- or bulking-dominated deformation mechanism.

Damage tolerance tests were also carried out on the ASNF aerogels from zero to 50% compressive strain. Supplementary Fig. 6e–g showed minor changes in the peak stress and small residual plastic strains (2%) even after 1000 cycles. These results demonstrated that ASNF aerogels could withstand large elastic deformation without accumulation of damage or undergoing any structural collapse. Large-scale samples often encounter issues of uniformity and mechanical property degradation. In this study, the large-scale samples (35.0 × 34.0 × 2.5 cm³) exhibited similar compression-recovery curves, thermal conductivity, and micromorphology at different parts, which

were almost identical to the small size samples tested in the following (Supplementary Fig. 7). These observations demonstrated that CIC processes could be potent for nanofiber aerogel formation because these materials could be easily scaled without alteration of the thermal insulating and mechanical properties of the resultant materials.

## Process intensification and structural evolution

Our CIC method decoupled the ice nucleation and growth processes compared with the traditional ice-template method. We confirmed the high-energy transfer efficiency of rotating freezing in preparing crushed ice through computational fluid dynamics (CFD) simulations[42]. The CFD models were constructed according to experimental setups, to better and timely present a thin liquid film and a vigorous flow generated on the drum surface (Fig. 1f, g and Supplementary Movie 3).

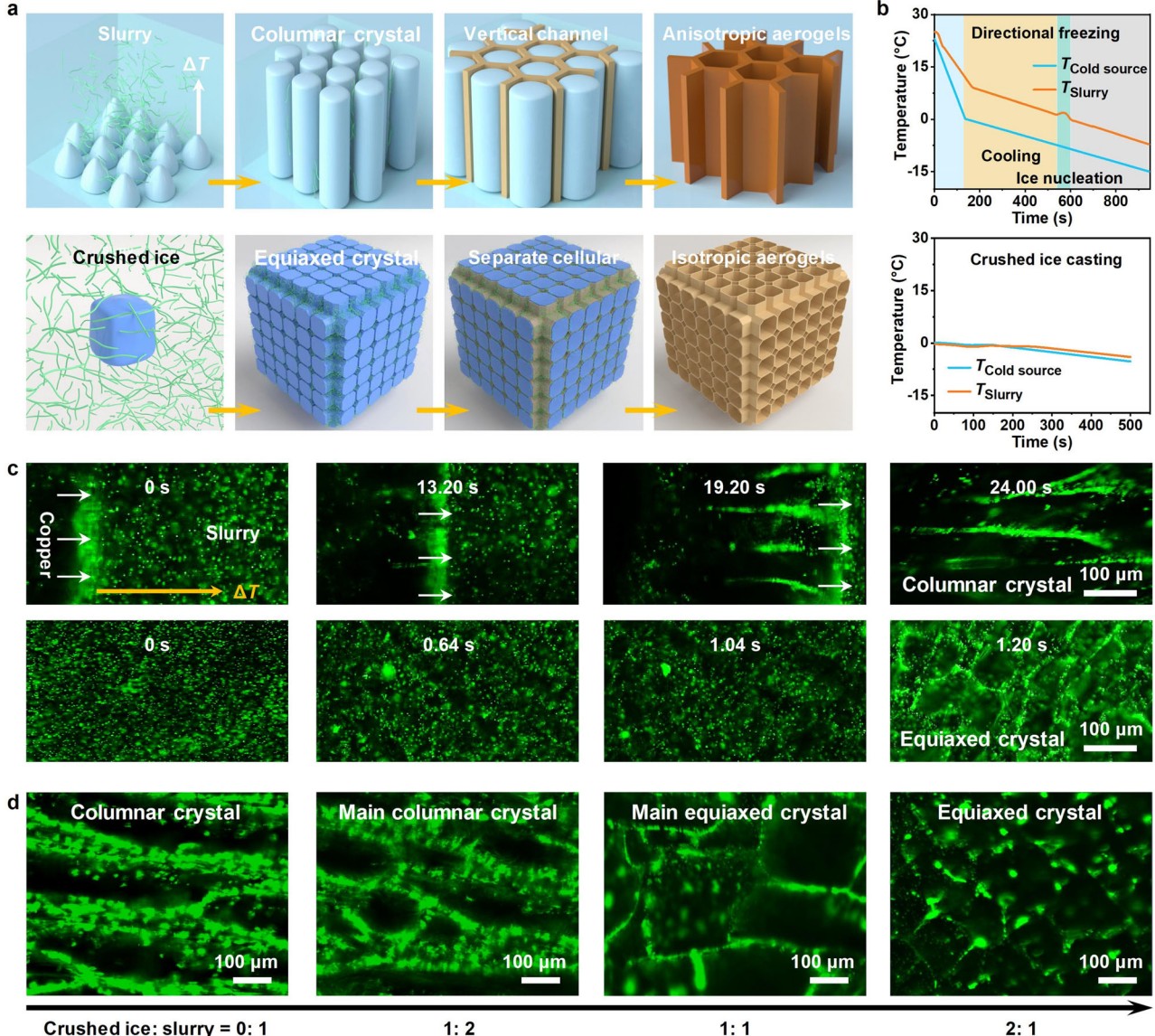

**Fig. 2 | Proposed freezing mechanism. a** Schematic illustration of the unidirectional freezing (upper) and CIC (bottom) processes. **b** Temperature variation over time during unidirectional freezing (upper) and CIC (bottom). **c** In situ observation on unidirectional freezing (upper) and CIC (bottom) using a optical-fluorescence microscope. **d** In situ observation on crushed ice casting using a optical-fluorescence microscope in different mass ratios for ice crystal to slurry.

Indeed, the heat transfer coefficient ($h$) on the RCD surface was always higher than the static state, corresponding to higher freezing efficiency. After working for 6.5 s, the $h$ of the rotating drum descended to a stable value of 56 W m$^{-2}$ K$^{-1}$. However, the $h$ on the static one reduced continuously owing to the increase of surface ice thickness and heat transfer resistance, which was consistent with the change in surface temperature (Supplementary Fig. 8a). This theoretically calculated result was further confirmed by the experimental results (Supplementary Fig. 8b). The RCD could produce 1 kg of crushed ice within 13.5 min at a constant yield. This production time is 2.8 times shorter than the plate freezing with the same surface area.

To obtain a bulk material, the re-casting of crushed ice is necessary. Typically, during unidirectional freezing, the ice crystals nucleate successively from the surface of the copper platform and grow preferentially in the vertical direction along the temperature gradient, resulting in a macroscopically long-range aligned lamellar distribution. In contrast, the freezing process regulated by artificially added ice crystals will show a completely different feature (Fig. 2a and Supplementary Movie 4). To probe the mechanism differences between

unidirectional freezing and CIC, we in situ observed the cooling process under an optical-fluorescence microscope after mixing with a small number of fluorescent polystyrene microspheres. Two comparative tests have been conducted at the same cooling rate of 1 °C min$^{-1}$ during freezing. In the unidirectional freezing process, the slurry temperature decreased linearly before ice nucleation. A sudden temperature rise could be observed when the slurry temperature reached 1.3 °C, which was attributed to the heat release upon nucleation. However, in CIC process, the setting temperature and slurry temperature were essentially consistent and no nucleation peak appeared, demonstrating the crystallization process does not require excessive external energy input to overcome the crystallization energy barrier (Fig. 2b).

Specifically, when the slurry was frozen on a cryogenic plate, ice nucleation and growth could expel nanofibers to the surface of the crystal column to form the aligned walls. The nanofibers formed a long-range lamellar structure and cast a solid anisotropic network. After 24 s, the cooling slope was almost identical to the initial state, indicating that the crystallization process has been completed. The

corresponding sequential top-view optical images further verified this orderly growth process (Fig. 2c, Supplementary Fig. 9 and Movie 5). However, owing to the no nucleation process in the CIC system, an ultrafast crystallization process of 1.2 s could be determined in the sequential optical and fluorescence images.

Next, we compared the overall freezing efficiency of different freezing methods. When the freezing distance was low, the CIC method had no significant advantage over directional freezing because two separate steps were required. However, when producing an ice block with a thickness of 3 cm, the total freezing time was 1.38 times shorter than that of the conventional unidirectional freezing process (Supplementary Fig. 10). With an increase in freezing distance, the freezing efficiency would be more significant. This can be explained as the process intensification from decoupling the ice nucleation and growth processes, including: (1) Thin film freezing process on rotating cryogenic drum can effectively increase heat transfer efficiency; and (2) Multi-point growth from 3D distribution of crystal nucleus can reducing crystallization path and accelerating the crystallization efficiency. Moreover, the whole process time from raw materials to products was also much lower than the traditional fiber felt/silica aerogel composites (Supplementary Table 2), demonstrating the practicality of this method.

In terms of resultant materials, the columnar and equiaxed crystals were the main components in ice blocks from unidirectional freezing and CIC, respectively, derived from the difference in the growth orientation. In the manufacturing process of industrial ingots, columnar crystals seriously affect the strength and toughness of materials. To increase the proportion of equiaxed grains, mechanical vibration or ultrasound inducting was applied to form more discrete nuclei. In the CIC process, ice crystals were artificially added to increase discrete nuclei, thereby promoting the formation of similar equiaxed crystals.

We investigated the influence of the ice-slurry ratio on the crystal structure of obtained ice blocks. When the proportion of ice crystals was small, most ice crystals would form oriented columnar crystals along the direction of the temperature gradient. Only a small portion of the area was affected by external ice crystals to form partial equiaxed crystals (Fig. 2d). When the proportion of ice crystals reached a moderate level (2:1), all pre-existing and 3D randomly distributed ice crystals could act as primitive growth sites. The adjacent crystals would unconstrainedly grow with no favorable location or orientations and ultimately form multi-domain bulk materials, along with the refinement of individual ice crystals. After the sublimation of ice crystals, the structure distributed along the ice crystal interface assembled by nanofibers was preserved. An isotropic aerogel with separate cellular networks could be readily obtained. The refined tortuous and complex pore structure would become an effective barrier for thermal, electronic, or acoustic transport.

## Mechanical and thermal insulation performances

Figure 3a showed the in situ SEM images of the material during the cyclic compression test (maximum strain up to 80%) (Supplementary Fig. 11 and Movie 6). The original sample showed an individual cellular microstructure with an average unit size of 100 μm, enclosed by thin walls (~2 μm thick). With the increase in compressive strains, the cell walls gradually bent and folded. Meanwhile, the cellular microstructure was densified. Upon unloading, the cell walls recovered fully to their original shapes, and so did the entire cellular microstructure. During one loading/unloading cycle, the material deformed elastically without any visible damage. These mechanisms exactly match the relative moduli–densities scale mentioned above.

To better understand the elastic deformation mechanism, we performed simulations using the finite element method (FEM) on a representative volume element (RVE) of our material. The cell walls were assumed to be isotropically linear hardening elastic-plastic material, whose Young's modulus was estimated as 78 MPa based on the model in literatures[43,44]. FEM simulations demonstrated the evolution of morphologies consistent with the in situ SEM experiments (Fig. 3b, Supplementary Fig. 12 and Movie 6). The walls buckled during the compression and then collapsed as the RVE was compacted. Owing to the thin cell walls, the FEM model showed no severe stress concentration, and the maximum value of von Mises stress did not exceed 10 MPa. Thus, the entire cellular structure retained its integrity and could spring back to its initial state after unloading. Furthermore, the stress *versus* strain curve predicted by FEM showed a high correlation with the experiments (Fig. 3c). After unloading, the stress-strain curve returned to its origin, which suggested that the material exhibited no residual deformation, as observed in the experiments. The entire curve during the loading-unloading cycle presented a notable hysteresis loop. The hysteresis loop area in our FEM simulation was smaller than in the experiments. This was probably because some energy dissipation mechanisms (such as the slippage and friction between nanofibers) were not considered in FEM.

Through introducing crushed ice, we successfully fabricated ASNF aerogels with separate cellular networks, enabling ultralight features. Figure 3d showed a piece of the ASNF aerogels with a volume of ~18 cm³, which could support on the tip of a dandelion ($\rho = 0.59$ mg cm⁻³). Besides, in contrast to reported unidirectionally freeze-dried materials[22], the resulting aerogels exhibited isotropic porous structure from the top- and side-views, which was proved by laser penetration experiment (Fig. 3e and Supplementary Fig. 13). A collimated 820-nm heat source with a spot size of 3 mm and an input power of 0.35 W was applied in directions perpendicular and parallel to the surface of the ASNF aerogels. The maximum temperature (150 °C) and heat transmission distance (8 mm) were identical in different directions. The macropore size distribution of ASNF aerogels was characterized by an automated mercury porosimeter (Fig. 3f). The average size of macropore in the ASNF aerogels was 39.86 μm and the porosity was estimated as 90.73%. We further investigated the mesopore size distribution. The lamellar ASNF sponges directly obtained from an SBS process and the anisotropic aerogels pre-fabricated using a unidirectionally freeze-drying method were adopted as comparisons[4,22]. Our aerogels had the highest Brunauer-Emmet-Teller (BET) specific surface area (21.89 m² g⁻¹) and Barrett–Joyner–Halenda (BJH) average pore sizes (4.86 nm), owing to the existence of commercial silica aerogel powders (Supplementary Fig. 14 and Table 3). However, the ASNF sponges showed a nearly untraceable specific surface area and mesopore volume.

We also compared the thermal conductivity of our ASNF aerogels with those of the lamellar sponges and anisotropic aerogels along the axial and radial directions (Fig. 3g). The lamellar sponges had a relatively high bulk density (~20 mg cm⁻³), corresponding to comparably higher solid conductivities than the other two aerogels. The low thermal conductivity of the axial direction was mainly provided by the thermal bridge inhibition effect, layer-by-layer air blocking effect, and the multilayer diffuse reflection effect[45,46]. However, in the radial direction, gas conduction and convection mainly contributed to thermal conductivity owing to the large gaps between lamellas, while the nanofiber walls serve as channels for solid conduction.

The thermal conductivities of anisotropic ASNF aerogels were measured to be about 0.026 and 0.035 W m⁻¹ K⁻¹ along the axial and radial directions, respectively. Anisotropic thermal properties of aerogel can enable efficient thermal dissipation along the axial direction, thus yielding enhanced thermal insulation in the radial direction. Thus, the thermal conductivity value in the axial direction is comparable to that of the reported silica nanofiber aerogels[10,22]. Moreover, the isotropic ASNF aerogels exhibited the lowest thermal conductivity of 0.020 W m⁻¹ K⁻¹ at 25 °C, modeled well by the effective medium theory (EMT) equation (see Supplementary Information for details)[26]. The reason can be explained as the columnar-equiaxed crystal transition in

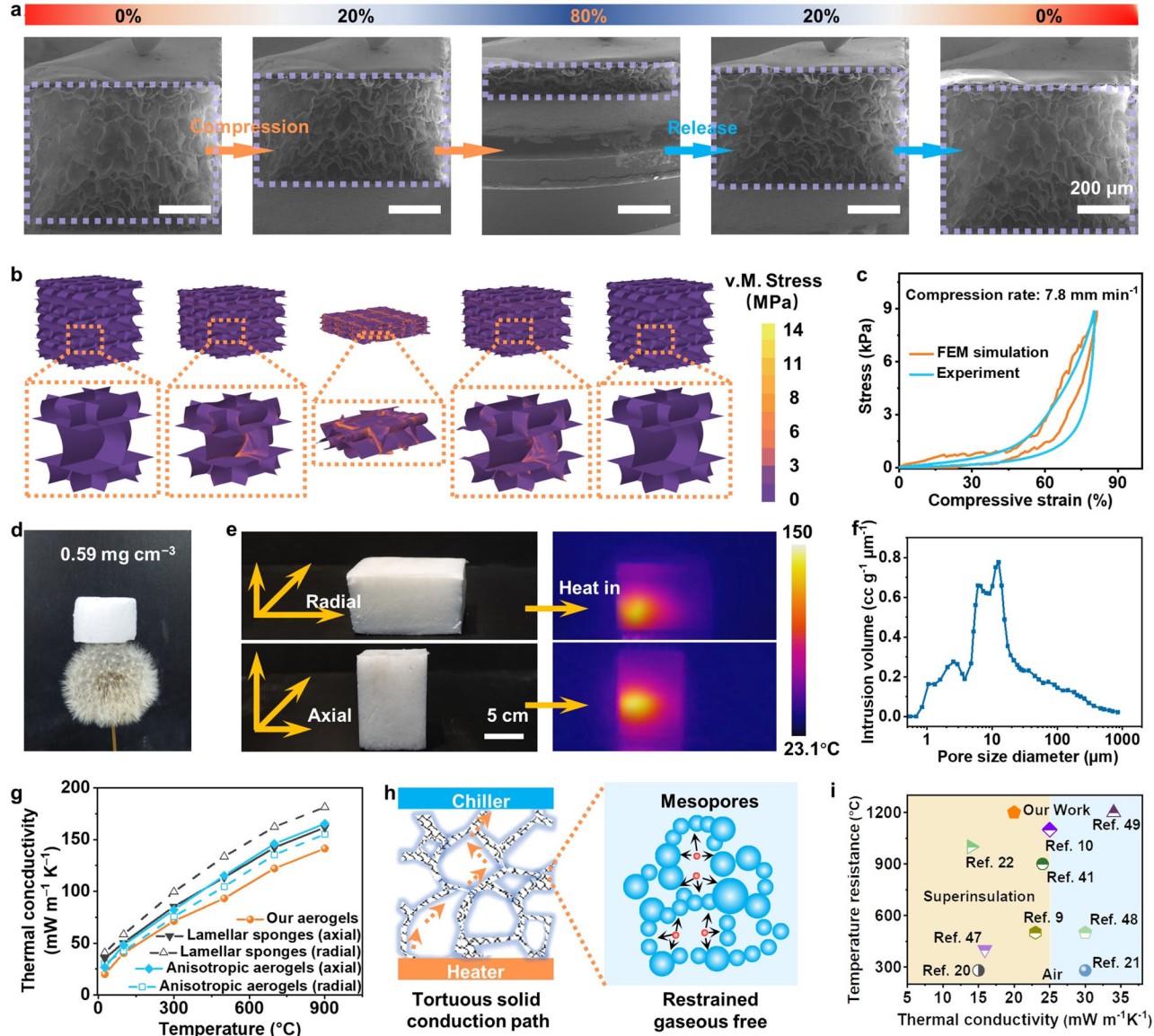

**Fig. 3 | Mechanical analysis and simulations. a, b** Structural evolutions of ASNF aerogels in a compression-release cycle and simulation of a nonlinear finite element model. **c** Compression-recovery profile comparison under 80% strain during experiment and simulation. **d** An optical image showing an 18 cm³ aerogel ($\rho$ = 0.59 mg cm⁻³) resting on the tip of a dandelion. **e** Infrared images of ASNF aerogels when illuminated by a laser from perpendicular and transection directions with a wavelength at 820 nm. **f** Pore size distribution of ASNF aerogels derived from automated mercury porosimeter. **g** Thermal conductivity versus temperature profiles of initial lamellar sponges obtained by SBS, anisotropic aerogels from directional freezing, and our aerogels. **h** Schematic of the heat transfer process in separate cellular ASNF aerogels. **i** Room temperature thermal conductivities in air versus temperature resistances for our aerogels and other previously reported ultralight materials. Numbers in the charts represent relevant references.

the freezing procedure resulting in thousands of repeating refined microscale units with a tortuous channel topology, reducing gaseous thermal conductivity and extending the length of solid heat conduction path.

We reconstructed the 3D structure of anisotropic ASNF aerogels and our aerogels from CIC technology using X-ray microtomography (Supplementary Fig. 15 and Movie 7). The microtomography images further confirmed the anisotropy of the aerogels from unidirectional freeze-drying with the straight, parallel aligned channels, while our isotropic aerogels consisted of several separate cellular networks. These macroporous structures composed of nanofibers built tortuous and complex solid conduction paths, slowing down the heat transfer efficiency (Fig. 3h). At the nanoscale, the abundant number of mesopores could restrain the free movement of molecules and significantly decrease the gaseous heat conduction (Knudsen effect)[26], resulting in

increased interfacial thermal resistance. At higher temperatures, radiative conduction increased correspondingly and the thermal conductivity of ASNF aerogels rose to around 0.125 W m⁻¹ K⁻¹ at 900 °C. In the testing temperature range, the thermal conductivity of the ASNF aerogels was always lower than those of the lamellar sponges and anisotropic aerogels. This suggested that the separate cellular structures also contributed significantly to the hindrance of thermal radiations owing to the multilayer diffuse reflection. Moreover, we compared the lowest thermal conductivity and highest temperature resistance of our ASNF aerogels with those of other typical thermal insulation materials in an oxidizing atmosphere, such as nanocellulose/graphene oxide[20], nanowood[21], SiO₂ aerogels[47], carbon nanotube aerogels[9], graphene/Al₂O₃[48], hBN aerogels[41], SiC@SiO₂ nanowire aerogels[22], SiO₂ nanofiber aerogels[10], and ceramic microfiber sponges[49] (Fig. 3i and Supplementary Table 4). The ASNF aerogels showed a

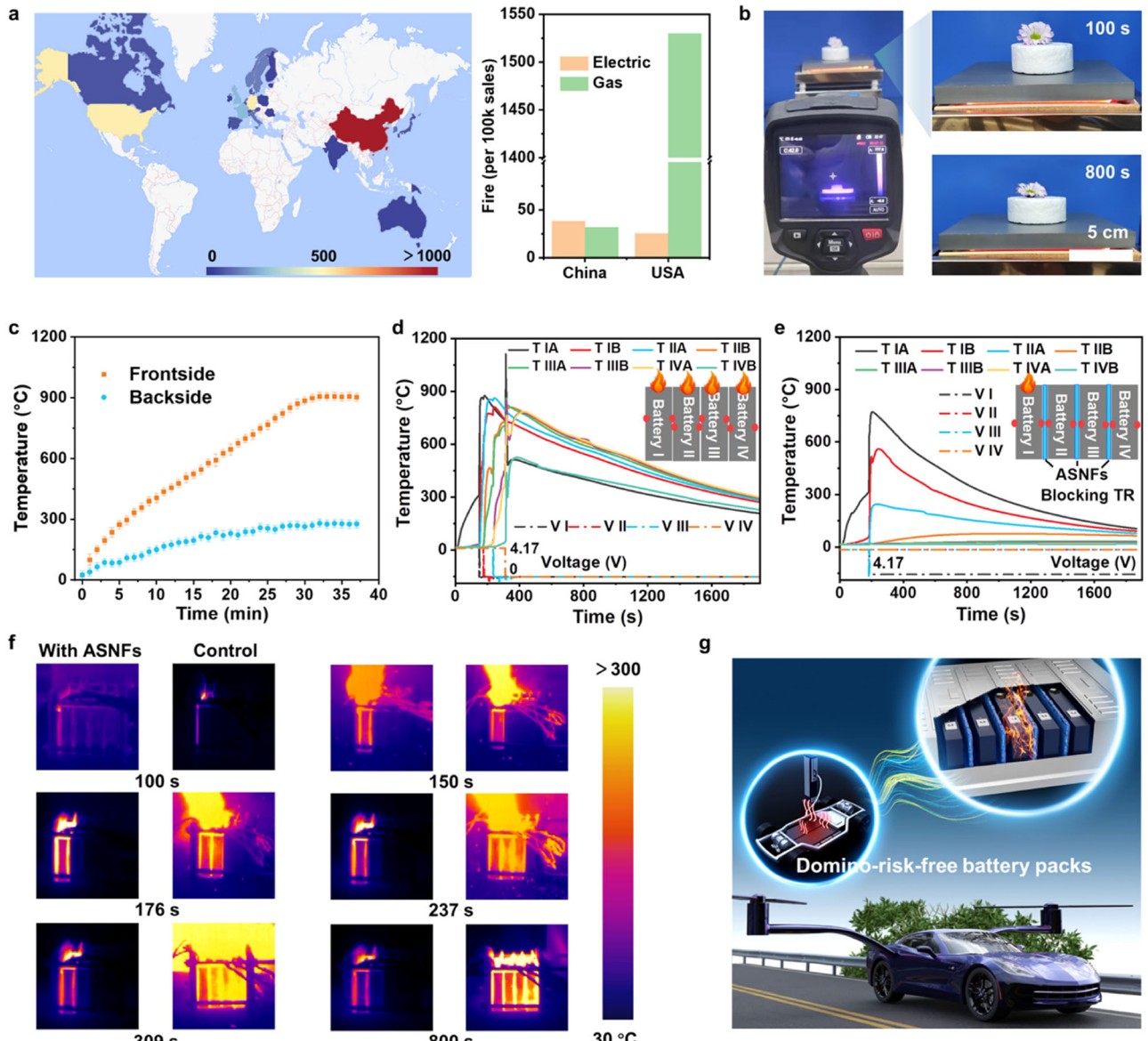

**Fig. 4 | Blocking of TR propagation in a battery module. a** Statistical chart depicting thermal runaway accidents of electric vehicles occurring globally in 2021. (Right) Comparison of electric and gas vehicle fire accident rates between China and the United States. **b** Set up for measurement of insulation property using a hot plate. **c** Time-dependent temperature profiles of the top and bottom planes of a 3-cm-thick aerogel placed on a hot plate at 900 °C. **d**, **e** Surface temporal evolution of NCM811 battery modules without and with the protection of ASNF aerogels during overheating-induced TR propagation tests. (Insets) Demonstration of corresponding module structures and temperature collection points. **f** Comparison of time-dependent images recorded during overheating-induced TR propagation tests. **g** Establish domino-risk-free battery modules and their applications in flying EV using ASNF aerogels.

lower thermal conductivity than most of the reported thermally insulating materials, well below that of standing air (0.025 W m$^{-1}$ K$^{-1}$). Furthermore, the long-term temperature resistance of ASNF aerogels was 1200 °C, which was the highest among the thermally superinsulating materials.

## Discussion

To fight against climate change and the fossil energy crisis, electric vehicles (EVs) and power stations equipped with high-energy-density lithium-ion batteries (LIBs) are gradually occupying the transportation and grid energy storage industries[50]. However, battery thermal runaway (TR) has become a major hurdle impeding practical applications[51–53]. In 2021 alone, more than 3000 and 500 field EV fire failures have been recorded in China and US, respectively, along with some sporadic accidents in other regions (Fig. 4a). Moreover, the EV fire rate of 0.04% has even surpassed that of gas-driven vehicles in

China. The EV fire usually initiates in a single battery and quickly spreads to the whole system like domino-toppling. The complex thermomechanical shock in battery fires with a LiNi$_{0.8}$Co$_{0.1}$Mn$_{0.1}$O$_2$ (NCM811) cathode has been considered as the most complicated hazard among other battery TR events[54]. Considering the low thermal conductivity and mechanical reliability of ASNF aerogels, the as-prepared materials can play an essential role in avoiding cascading TR failures in battery modules.

Most elastic structures with low thermal conductivity from polymers or carbonaceous materials cannot withstand high temperatures under ambient conditions. As ceramic families, ASNF aerogels are expected to possess temperature-invariant elasticity. Different viscoelastic properties, such as storage modulus, loss modulus, and damping ratio, were investigated over a broad temperature range of −100 to 500 °C at a constant frequency of 1 Hz. Indeed, the aerogels showed temperature-independent stable viscoelastic performances

and a consistently small damping ratio of ~0.1 (Supplementary Fig. 16a–c). A frequency dependency test (0.1 to 100 Hz) also showed stable viscoelastic properties over a wide temperature range of −100 to 500 °C. Furthermore, compression tests were conducted by exposing the materials to a butane flame (1300 °C) and submerging it in liquid nitrogen (−196 °C) (Supplementary Fig. 16d and Movie 8). The ASNF aerogels retained their resilience up to 80% compressive strain under extreme conditions. After several cycles, the aerogels fully recovered, with no obvious fracture. After a long-term high-temperature treatment (1200 °C for 24 h), the characteristic peak of mullite and cristobalite crystal phases appeared in the X-ray diffraction pattern (XRD) (Supplementary Fig. 17). With an average crystal size of 65.7 nm and crystallinity of 69% at this time, the nanograin-glassy dual-phase structure would not severely damage the strength and flexibility of ASNF aerogels[4]. However, we could observe many defects formed on the nanofiber surface after calcination at 1300 °C; along with the decrease of elasticity and the increase of density, the nanofibers would gradually become brittle and finally be crushed.

The time-dependent thermal insulating capacity of ASNF aerogels was evaluated by recording the infrared images of the materials placed on a heating SiC stage at 900 °C (Fig. 4b, c, Supplementary Fig. 18 and Movie 9). The ASNF aerogels showed a slow temperature increase on the top side up to a maximum temperature of around 270 °C after 30 min, establishing a long-term stable temperature gap of 620 °C. We further evaluated the domino-risk mitigation capacity in a high-energy-density battery module by recording the TR propagation process between single batteries. As shown in Fig. 4d, a practical four-cell NCM811 module with a cumulative energy density of 512 Ah was heated laterally to induce a TR process. In the unprotected module, the surface temperature of the TR battery sharply rose from 320 to 861 °C in 4 s, accompanied by voltage vanish, battery swelling, smoke venting (high-speed airflow blowing), sparking, fire jetting, and sustained combustion (Supplementary Fig. 19 and Movie 10). Then, the TR events rapidly propagated from the initiating battery to its neighbors, consequently leading to disastrous failure at the system level. Our study measured the average TR propagation time between adjacent cells as 162.4 s. All batteries shared high maximal temperatures (~875 °C) with similar temperature response profiles. Moreover, this TR propagation caused an explosion accident on the fourth battery owing to the accumulation of energy, leading to the module jumping and violent fire-spitting. The explosion event could be further evidenced by the molten droplets and holes on the metal shell.

In stark contrast, our ASNF aerogels could completely block the TR propagation without additional assistants and specific energy loss. The volumetric energy density of the module was determined as ~677.6 Wh l$^{-1}$ in the initial state and ~589.3 Wh l$^{-1}$ with ASNF aerogels. We investigated the temperature variation and open-circuit voltage (OCV) responses during TR events, as shown in Fig. 4e, Supplementary Fig. 20 and Movie 11. After inducing TR by over-heating, the highest thermal shock power of 215 kW (241 °C s$^{-1}$) could be calculated based on the temperature response of the trigger battery (Supplementary Fig. 21), which is much higher than other battery systems, such as NCM523 battery (55.27 kW)[4]. However, the OCVs of the adjacent battery remained stable, the highest temperature gap of 315 °C could be established by ASNF aerogels, and the surface temperature of the neighbors did not reach the TR trigging temperature (~200 °C)[53]. The infrared images further demonstrated this process (Fig. 4f).

The ASNF aerogels exhibit an array of exceptional behaviors that are highly desirable for lightweight domino-risk-free battery modules toward air transportation and aerospace exploration where safety, weight savings and energy density are critical considerations (Fig. 4g). The ASNF aerogels have high mechanical properties, high-temperature resistance, low density, and low conductivity, realizing the TR blocking of the NCM811 LIB systems. These performances exceed the most widely used or reported firewall materials such as perlite, glass fiber,

rock wool panel, calcium silicate, silica aerogel, asbestos[55], and porous silica voxels[56]. Moreover, the ASNF aerogels with an isotropic structure could be trimmed into arbitrary shapes to fit the needs of the current diversified battery or module forms (Supplementary Fig. 22).

In summary, we demonstrated a straightforward concept of combing superelasticity and thermal superinsulating by fabricating separate cellular structured lightweight ceramic nanofibrous aerogels. The CIC process could create thousands of individual microscale units that enable an ultralow thermal conductivity of 0.020 W m$^{-1}$ K$^{-1}$ and a considerable recovery from deformation at 90% strain. The fatigue test also showed no significant damage after 1000 compression cycles. Moreover, oxide ceramic constituents guaranteed the aerogels with high thermal and chemical stabilities in an oxygen-containing high-temperature environment. A 5 mm thick aerogel could completely block the devastating TR propagation within a 212 Ah LIB module with NCM811 cathode. This aerogel can help alleviate the safety challenges related to flying devices or grid energy storage systems, and stimulate the development of advanced lightweight battery systems with high-security standards.

## Methods
### Fabrication of ASNF aerogels
ASNF aerogels were 3D constructed from flexible alumina silicate nanofibers, CNFs, commercial silica aerogel powers (SAPs), and inorganic binders based on CIC technology and freeze-drying. In total, 1 g alumina silicate nanofibers, 0.1 g SAPs (particle size ~200 nm) and 0.1 g CNFs were dispersed in 200 g deionized water to prepare uniform nanofiber dispersions using a fiber disintegrator with a rotating speed of 50 rad s$^{-1}$. Then, 7.65 g Si-B sol was added to the nanofiber suspensions and gently stirred for 30 min, which was synthesized by mixing and hydrating 0.488 g TEOS, 0.017 g boric acid, 4.88 g ethanol, and 5 mg $H_3PO_4$ in 2.26 g deionized water. The solution was rapidly transformed into nanofiber crushed ice through the crushed-ice-making machine at a rotating speed of 50 rad min$^{-1}$. The cryogenic drum has a diameter of 13 cm and a width of 11 cm. The gap between the scraper and the drum is 3 mm. The obtained nanofiber crushed ice was mixed with nanofiber suspensions (normal temperature) in a mass ratio of 5:1 to eliminate the gap between crushed ice. For typical experiments, a 200 g ice-dispersion mixture can be obtained after a half-minute blending using a thermally insulated mixing tank and customized agitator paddles (Supplementary Fig. 22). Next, they were transferred to the desired mold, frozen on a cold plate in a low-temperature chamber (−20 °C), and then freeze-dried for 18 h to obtain the ASNF aerogel precursors. Owing to the ice-dispersion mixture having a certain fluidity, it can fill the mold without leaving pores. Following, the ASNF aerogels precursors were calcined at 600 °C in a muffle furnace for 30 min to produce the pure inorganic ASNF aerogels. Meanwhile, the bulk density of the form could be arbitrarily adjusted in the range of 0.59 to 20 mg cm$^{-3}$ by changing the concentration of alumina silicate nanofibers. The large-scale samples were freeze-dried in an industrial vacuum freeze dryer. When comparing the ice block production rate of the conventional unidirectional freezing to CIC, the same mold size (18 × 18 cm$^2$ in-plane) and freezing environment (−20 °C) was adopted. Except as noted, all the structure and property investigations were performed by the form with a density of 5 mg cm$^{-3}$. The preparation process of alumina silicate nanofibers and other aerogels can be seen in the Supplementary Materials.

### Characterization
The morphology was investigated using SEM (Merlin). The structure feathers were characterized by thermogravimetric analysis (Mettler Toledo STA409PC/DIL 402PC), Fourier transform infrared (FT-IR) spectroscopy (NICOLET IS50), XRD (D8 ADVANCE), and X-ray photoelectron spectroscopy (XPS, PHI 5000C ESCA). Compression tests were conducted using a DMA instrument (TA-Q850) equipment with

an environmental chamber. The sample diameter is 22 mm, and its height is 7.8 mm. The microscopic architecture evolution of the ceramic nanofibrous aerogels was characterized by focused-ion-beam-SEM (Crossbeam 340) equipped with a nano-manipulator. The rapid recovery of a ceramic nanofiber aerogel was recorded using a high-speed HUAWEI P30pro camera operated at 960 frames per second. The thermal conductivity was measured using Hot Disk TPS3500.

When in situ observed the freezing process induced by a copper platform under an optical-fluorescence microscope (ZEISS LSM 900), the slurry with 0.5% of nanofiber and 1 mg ml$^{-1}$ green fluorescent polystyrene microspheres was poured into the quartz mold. A copper block was placed on one side of the mold at a cooling rate of 1 °C min$^{-1}$ during observation. In terms of SC processes, the designed snow-slurry mixture was frozen at the same cooling rate to generate a proper ice growth velocity for better observation.

The Euler-Lagrange method was adopted to simulate the heat transfer process between the drum and liquid film using the solidification and mining model. The grid was made up of quadrangle segments with a number of 145,000. The Realizable $k$-$\varepsilon$ model was chosen as the turbulence equation. The parameters of the drum and liquid film used in Fluent were listed in Supplementary Table 5. To speed up the convergence process, a simple algorithm with a pressure solver was used. To better capture the temperature change process, the time step was set as 0.01 s. The calculation converged when the energy residual curve was lower than $1 \times 10^{-6}$, and other residual curves were lower than $1 \times 10^{-3}$.

Simulations of the compression test were carried out using the FEM with the commercial software Abaqus/Explicit R2017. The material was simplified as an RVE containing $3 \times 9$ units cells shown in Supplementary Fig. 12. The unit cell ($200 \times 200 \times 200\ \mu m^3$ in size) was constructed by 2 μm thick walls. The cellular walls in the actual aerogels were networks consisting of randomly aligned ASNFs. In the FEM simulations, the ASNF networks were homogenized as continuous thin shells that behave as an isotropic linear elastically. For thin shells consisting of randomly aligned fibers, the equivalent Young's modulus for the homogenized shell can be estimated by[43,44]:

$$E_{\text{eff}} = \frac{E_{\text{bulk}}\left(\frac{\pi n l^2}{4\lambda \hat{t}} - \frac{2.00 + 2.44/\hat{t}}{\lambda}\right)}{6\left(\frac{2.86 + 2.5/\hat{t}}{\lambda^{\frac{3}{4}}} - \frac{2.00 + 2.44/\hat{t}}{\lambda}\right)} \quad (1)$$

in which $\hat{t} = 10$ is the ratio of the shell thickness over the fiber diameter, $n = 0.02\ \mu m^{-2}$ is the average fiber number per unit area, $\lambda = 50$ is the aspect ratio of ASNFs, $E_{\text{bulk}} = 73$ GPa is Young's modulus of bulk ceramics. Then, Eq. (1) estimated the effective elastic modulus of the homogenized walls ($E_{\text{eff}} = 78$ MPa). The FEM model was discretized using the 4-node linear thin shell element (S4R) with a mean mesh size of $2 \times 2\ \mu m^2$. Periodic conditions were in X and Y directions. A rigid plate was set beneath the cellular model and fixed. Another rigid plate right on the top of the model moved down at the speed of 7.8 mm min$^{-1}$ until an 80% strain, and returned to its original position at the same rate. During the load/unload cycle, frictionless contact was set between all surface pairs.

**Battery thermal runaway propagation tests**
Four fully-charged 53Ah LIBs, using NCM811/graphite as electrodes, formed a battery module using the same pre-clamp of 1 N m in an overheating-induced TR propagation test. To avoid excessive compression of the aerogels, four mica gaskets (10 mm in diameter and 5 mm in height) were pre-bonded on four corners of the battery. Batteries I, II, III, and IV orient the batteries toward the direction where the heating plate (1 kW) was directly connected to the battery I. Three ASNF aerogels were placed between batteries, while four batteries

were not in direct contact with other metal holders or experimental chambers. Eight micro-thermocouples (T1–T8) were firmly attached to both sides of each battery to measure the surface temperature. Data recorders were used to record the temperature and voltage of each battery. Digital and IR cameras were employed to monitor the experiment process.

## Data availability
The data supporting the findings of this study are included within this article and its Supplementary files. Any additional information is available from the authors upon request. Source data are provided with this paper.

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

## Acknowledgements

This work is jointly supported by the Basic Science Center Program of the National Natural Science Foundation of China (NSFC) grant No. 52388201, and NSFC Grants Nos. 52325312, 52302512, and U21A20170, Beijing Natural Science Foundation Grant No. JQ19005, and China Postdoctoral Science Foundation Grant No. 2021M691713.

## Author contributions

W.L., X.W., J.Z., and H.W. contribute to the conceptualization. L.L., Y.Z., Y.G., X.F., F.Z., and B.Z. contribute to the methodology. L.L., Y.Z., Y.G., F.Z., Z.T., P.F., and M.Z. contribute to the investigation. L.L., Y.G., and H.N. contribute to the visualization. X.F., B.Z., S.Z., X.W., J.Z., and H.W. supervise this project. L.L., Y.Z., and Y.G. contribute to the writing—original draft. X.W., J.Z., and H.W. contribute to the writing—review and editing.

## Competing interests

The authors declare no competing interests.

## Additional information

[1]State Key Laboratory of New Ceramics and Fine Processing, School of Materials Science and Engineering, Tsinghua University, 100084 Beijing, China. [2]National Engineering Research Center of Electric Vehicles, Beijing Institute of Technology, 100081 Beijing, China. [3]State Key Laboratory for Turbulence and Complex System, Department of Mechanics and Engineering Science, College of Engineering, Beijing Innovation Center for Engineering Science and Advanced Technology, Peking University, 100871 Beijing, China. [4]State Key Laboratory of Automotive Safety and Energy, Tsinghua University, 100084 Beijing, China. [5]School of Chemistry and Chemical Engineering, North University of China, Taiyuan 030051, China. [6]National Laboratory of Solid State Microstructures, College of Engineering and Applied Sciences, Jiangsu Key Laboratory of Artificial Functional Materials and Collaborative Innovation Center of Advanced Microstructures, Nanjing University, Nanjing 210008, China. [7]Guangdong Huitian Aerospace Technology Co., Ltd, Guangzhou 510006, China. [8]Laboratory for Building Energy Materials and Components, Swiss Federal Laboratories for Materials Science and Technology, Empa, 8600 Dübendorf, Switzerland. [9]These authors contributed equally: Lei Li, Yiqian Zhou, Yang Gao. ✉e-mail: liweiwei197@126.com; xdwei@pku.edu.cn; jiazhu@nju.edu.cn; huiwu@tsinghua.edu.cn

