## [Peer Review File · Nature Communications]

Large-scale assembly of isotropic nanofiber aerogels based on columnar-equiaxed crystal transitionREVIEWER COMMENTS

Reviewer #1 (Remarks to the Author):

This manuscript presents a new concept of snow-casting (SC) technique, which is combined with a continuous rotating ice crystallization (RIC) technology, to solve the problem of non-uniform temperature gradient occurring in the conventional freeze casting method. This work also demonstrates the feasibility of SC method to produce ultralight, large-scale, thermal superinsulating (0.020 W/m K) and flexible Al₂O₃-SiO₂ nanofibers aerogels (ASNAs), which can find potential applications in aircraft, unmanned aerial vehicles and flying cars. In addition, the ASNAs are claimed to have potential to enhance the battery safety by blocking the thermal runaway in modules when used as separator between cells. The concept of SC also offers high flexibility of the shape of final products, expanding the range of applications. The manuscript has sufficient overall novelty suitable for publication in the Journal. However, there are several critical issues that need to be addressed before it can be reconsidered for publication.

1. The paper structure is not well-organized in terms of division. It is rather difficult to follow due to the absence of section headings.
2. The authors should state the freezing temperature used in the RIC process.
3. The time taken for ice crystal formation using SC is compared with that using unidirectional freeze-casting (FC). Because both the freezing temperatures used and the microstructures produced from these two methods are different, the authors should justify the choice of unidirectional FC for comparison. In addition, the unidirectional FC technique produces frozen samples in the final shape in one-step from the solution, whereas the SC involves an RIC process to prepare snowflakes and second freeze-casting in a cryogenic environment where the nanofiber suspensions are mixed with snowflakes to eliminate the gap between snowflakes. The entire fabrication times should also be compared rather than the time for ice nucleation alone. Essentially, it is difficult to make a direct comparison between the two techniques as each of these techniques stands alone with different approaches.
4. Associated with Comment #3: Instead of comparing the fabrication time between SC and unidirectional FC techniques, the authors may also consider comparing the fabrication time of SC with other fabrication techniques for large-scale silica aerogel blankets, because they are of an isotropic structure and commercially available.
5. Fig. 4d presents a digital image showing an ultralight feature of ASNAs with an ultralow density of 0.59 mg/cm³. Except this image, however, all other properties and structures shown are those with a much higher density of 5 mg/cm³, as stated in Method (p.14 Line 322-323). This information should be included in the main text, instead of in Methods, to avoid any misunderstanding.
6. Associated with comment 5: The ultralight weight is an important consideration for aviation applications as proposed. As such, the authors should justify the use of ASNAs with 5mg/cm³ for studies of the properties and structures. The optimal concentration of Al₂O₃-SiO₂ nanofibers and the final density of aerogels should also be discussed in detail.
7. Fig. 4f: Only one thermal conductivity is recorded for each anisotropic aerogel (prepared from the SBS and directional freeze-casting processes). Report the thermal conductivities in both the alignment and transverse directions for anisotropic aerogels.
8. According to SI Section 6 and Methods, apart from the fabrication method, the composition and thermal treatment of aerogels are not identical. Therefore, the validity of thermal conductivity comparison between different fabrication methods (Fig. 4f) is questionable. The difference in composition between different samples should be clearly declared. In addition, the use of thermal treatments for directionally freeze-cast aerogels and ASNAs while no treatment for lamellar sponges should be fully justified.
9. Minor comment: In Page 10 Line 214, the unit for bulk density should be mg cm⁻³.

Reviewer #2 (Remarks to the Author):

Review of « Snow casting of large-scale isotropic nanofiber aerogel »

The authors expose in this manuscript a variation of ice-templating enabling the production of large (m² range) pieces, with isotropic porous microstructures. The applicability of the process is then demonstrated with different materials (proof of concept).

The novelty of the process is its division into two freezing steps:

- a first step where only small bits of suspension (or solution) are ice-templated and scrapped from the cold surface repetitively. This ensures the relatively rapid production of a large volume of frozen bits (which the authors refer to as « snow »)
- a second step where the « snow » is mixed with additional native suspension (to fill the gaps between the snow bits), and frozen, before being freeze-dried.

Since the novelty is in the process itself, I would have expected a deep, parametric investigation of the process and the influence of the various parameters, that would have provided a good overview and understanding of the process, its possibilities, control, and limitations. Instead, only a small part of the paper (approx 20%) is devoted to the process, and almost no informations and data are provided about its control and properties. The majority of the paper is about the proof-of-concepts materials obtained with this process. Although it's nice to demonstrate what the process can do, this poor balance makes the paper confusing and a bit shallow. Examples of these points are discussed below.

The paper is also filled with exaggerated and hype statements which I found annoying (« excellent », « outstanding », etc.) and not very scientific. Great science does not need hype.

Specific comments:

- the term « snow » is fancy but misleading. Snow crystals are grown in very different conditions with a different physics (solid/vapor equilibrium). I think the authors should use a different term.
- line 59-60: « it is challenging to scale up the thick bulk material fabrication while maintaining consistent properties ». This challenge is supposedly solved here but actually not demonstrated. No mention is made of the maximum thickness obtained, for example. From the figure 1b and f, it seems that the thickness is of ~7-8cm (?), which is about twice larger than what has been demonstrated by conventional freeze casting (pieces several cm thick have already been reported). What is the maximum thickness achieved here? Scaling up the lateral dimensions is not so challenging (several papers have already been published on this), the challenge in scale-up so far has always been the increase in thickness, which is not clearly reported and discussed here. From what is exposed here, I suspect it is actually not so simple, since the « snow » is mixed with additional suspension into a block that needs to be frozen completely, as in conventional freeze-casting. These limits are not discussed in the manuscript.
- l72: « these building blocks often results in interconnected macro-scale open pores, whose dimensions are greater than the standard free path of air ». The authors imply here (and later in the paper) that their approach is better in this regards. However:
 - no pore characterization and pore size distribution is reported here
 - the SEM pictures reveal that the porous structure is clearly macroporous (micro- or mesopores could also be present but not visible on the SEM pics) and interconnected.I thus don't see anything unique or specific in these structures. Such strong claims should be supported by experimental data, which is not the case here.
- line 87: the « snow » is mixed with additional suspension and cast into blocks that need to be frozen completely. Is the additional suspension cooled before being mixed with the snow ? Is the mix homogenized? Are there any precautions to take to avoid thawing? How much pressure is applied to packed the mix and ensure no pores are left? No mention is made of how long it takes to freeze such large blocks. No mention is made either about the required freeze-drying time. Too little informations about this important step are provided in the manuscript.
- line 89: « the SC process was was easily scaled in a linear fashion to over 1.2m² area without alteration ». Where are the data supporting this statement? I only see a (nice) picture of a very large piece, but I don't see a comparison of the microstructure/properties for pieces of increasingly large dimensions.

- line 100 (and other places): many claims are made about the supposed advantages of the process, but these advantages are not really discussed, and no mention is made of the limits. For example, the SC process is more complex (more steps) than the standard freeze-casting process. The first freezing step may be faster, but a second freezing stage is required. Overall, it is very likely that the overall complexity and thus cost of the process is at least similar if not greater than conventional freeze-casting.
- line 104 « This was noteworthy as SC technology was incompatible with slurry-forming technology ». It don't understand this statement, could you please elaborate?
- line 122: « in a snow slurry system, the ice crystal tend to nucleate simultaneously ». If I understood correctly the process, the « snow » scrapped from the rotating drum is collecting and packed. The ice crystals are thus already present in the snow and not thawed at this stage. The ice crystals in the snow thus continue growing, there is no need to nucleate novel crystals.
- line 127: « we sucessfully realize complex architectures in snow-casted materials ». It's not particularly complex, it's either random (and somewhat isotropic), or already reported before (e.g. Fig S6).
- line 136 « this structure exhibited a special micro-orientation and macro-isotropic nature ». I don't see what's special (what does « macro-isotropic » means?) here, more explanations should be provided. The structure shown in Fig S6 is typical of freeze-casted fibrous structures (e.g. freeze-casted carbon nanotubes structures), which has been reported in numerous papers before. It's nice but I don't see anyting special or novel here.
- Fig 2D: I do not understand the figure. What is the direction of the temperature gradient or sample with respect to the surface?
- Fig 3D: as far as I can tell, this not a standard test. It is thus not possible to compare the behavior with other materials.
- line 320 « the bulk density [...] could be arbitrarily adjusted in the range of 0.59 to 20 mg/cm³ ». Where are the data? What's the variability? How reproducible is the process?

I am not confident to evaluate the modelling part of the specific materials properties reported here.

Overall, the paper is not focussed. Too little details are provided about the novel process and its control and limits, and too many informations are provided about a myriad of very different materials. This variation of freeze-casting appears thus promising but too little informations are provided about the process at this stage to make it a convincing, real advancement.

Reviewer #3 (Remarks to the Author):

The authors describe the methodology and successful synthesis of an ultralight, large-scale, thermal super insulating, and flexible nanofiber aerogel using a continuous rotating ice crystallization. The team also demonstrated the capability of a 5 mm thick film to decrease the risk of thermal propagation, which is a major challenge with lithium-ion batteries and an area of the field with a continuous need of advancements.

I recommend the authors to discuss the challenges and trade-offs of mitigating the risk of propagation and the added weight. What increase in parasitic mass and volume ratios do the nanofiber aerogel film introduce to the cell stack? Parasitic mass ratio is defined as mass of cell stack divided by mass of cells only. This may show that adding thermal capacitance to the cell stack is not an effective mitigation strategy for propagation resistance.

I recommend the paper for publication.

According to these comments and suggestions from three referees, we have carefully revised the manuscript with all the changes highlighted. The comments are reproduced and our responses are given directly in different color (blue). The point-by-point responses to Reviewer #1, Reviewer #2, and Reviewer #3 are listed in the following pages. All page numbers refer to the revised manuscript file with tracked changes.

Responses to Referee #1

This manuscript presents a new concept of snow-casting (SC) technique, which is combined with a continuous rotating ice crystallization (RIC) technology, to solve the problem of non-uniform temperature gradient occurring in the conventional freeze casting method. This work also demonstrates the feasibility of SC method to produce ultralight, large-scale, thermal super-insulating ($0.020 \text{ W/m}\cdot\text{K}$) and flexible $\text{Al}_2\text{O}_3\cdot\text{SiO}_2$ nanofibers aerogels (ASNAs), which can find potential applications in aircraft, unmanned aerial vehicles and flying cars. In addition, the ASNAs are claimed to have potential to enhance the battery safety by blocking the thermal runaway in modules when used as separator between cells. The concept of SC also offers high flexibility of the shape of final products, expanding the range of applications. The manuscript has sufficient overall novelty suitable for publication in the Journal. However, there are several critical issues that need to be addressed before it can be reconsidered for publication.

Response summary: We thank the reviewer for the positive comments on our manuscript. These comments were very helpful in improving the quality of manuscript. We have revised the manuscript according to the reviewer's suggestions. Moreover, we have carefully addressed, point by point, all the comments the reviewer has raised. We hope that the reviewer will find our responses satisfactory and convincing. Our responses to the reviewer's comments are listed below.

Comment 1: The paper structure is not well-organized in terms of division. It is rather difficult to follow due to the absence of section headings.

Response: We are thankful for the constructive comments. Indeed, the frame structure of the paper is not well-organized in the previous version. According to the referee's suggestions, we have restructured the frame of the article, disentangle the logical order, and added the section headings in the revised manuscript. After adjusting, the manuscript has been improved a lot.

Revise details:

- Page 3, Line 29, added "Fabrication of isotropic nanofiber aerogels."
- Page 6, Line 1, added "Observation of the freezing process."
- Page 7, Line 25, added "Temperature-invariant mechanical performances."
- Page 10, Line 15, added "Mechanical analysis and simulations."
- Page 12, Line 4, added "Thermal insulation performance."
- Page 13, Line 26, added "Proof-of-concept experiments."

Comment 2: The authors should state the freezing temperature used in the RIC process.

Response: Thank you very much for your valuable comments. We have added the freezing temperature in the article.

Revise details:

- Page 3, Line 4, the "The nanofiber was then dispersed in a suitable solution and rapidly frozen on the surface of a rotating cryogenic drum (RCD) at -20 °C and scraped into crushed ice (Supplementary Fig. 2 and Video 1, see Supplementary Materials for details).".

Comment 3: The time taken for ice crystal formation using SC is compared with that using unidirectional freeze-casting (FC). Because both the freezing temperatures used and the microstructures produced from these two methods are different, the authors should justify the choice of unidirectional FC for comparison. In addition, the unidirectional FC technique produces frozen samples in the final shape in one-step from the solution, whereas the SC involves a RIC process to prepare snowflakes and second freeze-casting in a cryogenic environment where the nanofiber suspensions are mixed with snowflakes to eliminate the gap between snowflakes. The entire fabrication times should also be compared rather than the time for ice nucleation alone. Essentially, it is difficult to make a direct comparison between the two techniques as each of these techniques stands alone with different approaches.

Response: Thank you very much for your insightful comments and suggestions. In general, nondirectional freeze-casting is a common method for preparing huge ice block. However, the nondirectional freezing process usually encumbered by the great nucleation resistance and low heat transfer capacity owing to the relatively low thermal conductivity of ice. We often observe phenomena of supercooled water (liquid water below 0 °C) and hollow ice block (center is liquid). This process is time-consuming and inefficient (Fig. 1e), and has been replaced by the induction nucleation of metal materials, which can also be called unidirectional freezing.

In unidirectional freezing process, the heat conduction is in the axial direction and the heat is barely not conductive in the vertical direction. Specifically, when the slurry was frozen on a cryogenic plate, ice nucleation and growth can expel nanofibers to the surface of crystal column to form the aligned walls. The nanofibers were then formed a long-range lamellar structure and cast a solid anisotropic network. This method combined with formation mechanism and regulation strategies has been widely reported in various literatures¹⁻¹². At the experimental level, liquid nitrogen is usually used to fabricated the aerogels. However, once expanded to largescale samples, the controllability and cost of this method will face huge challenges.

As the reviewer noted, we compared the ice block production rate between the conventional unidirectional (one step) and directional freezing (one step) with crushed ice casting (two step). The same mold size (18×18 cm² in-plane) and freezing environment (−20°C) were adopted. The total required freezing time is 1.38 times

shorter than that of conventional directional freezing process, and 1.92 times shorter than that of unidirectional freezing process, when producing 1.2 kg ice block (Fig. 1e and Supplementary Fig. 5).

Fig. R1. (Left) Comparison of ice block production rate between the conventional directional and nondirectional freezing with crushed ice casting. **(Right)** Production rate of crushed ice manufacturing and crushed ice re-casting steps.

Revise details:

- Page 5, Line 20, added “The total required freezing time is 1.38 times shorter than that of conventional directional freezing process, and 1.92 times shorter than that of nondirectional freezing process, when producing 1.2 kg ice block (Fig. 1e and Supplementary Fig. 5).”
- Page 17, Line 9, added in Methods, “When comparing the ice block production rate of the conventional unidirectional and directional freezing to crushed ice casting, the same mold size (18×18 cm² in-plane) and freezing environment (−20 °C) were adopted.”

Comment 4: Associated with Comment #3: Instead of comparing the fabrication time between SC and unidirectional FC techniques, the authors may also consider comparing the fabrication time of SC with other fabrication techniques for large-scale silica aerogel blankets, because they are of an isotropic structure and commercially available.

Response: Thank you very much for your constructive suggestions. As suggested by the reviewer, we have compared the fabrication time of isotropic ASNf aerogels with traditional fiber felt/silica aerogel composites. As shown in Supplement Table 1, at

present, large size fiber felt/silica aerogel composites were usually prepared by spraying silica gel solution on fiber in industry. The gel solution is prepared by sol-gel method for about 24.5 h. Secondly, the aging and modification-replacement time of gel solution after spraying is about 48 h. Finally, the final aerogel felt is obtained after drying (~24 h). However, the whole process time from raw materials to products of our crushed ice casting was about 72.6 h. Therefore, compared with the traditional sol-gel method, the crushed ice casting method has significantly advantaged in improving the production efficiency.

Supplementary Table 1. Whole process time from raw materials to products comprison between isotropic ASNF aerogels and the traditional fiber felt/silica aerogel composites (Use $300 \times 300 \times 10 \text{ mm}^3$ sample as standard).

Astropic ASNF aerogels*		Fiber felt/silica aerogel composites**	
Procedures	Time (h)	Procedures	Time (h)
Nanofiber preparation	~24.0	Gel preparation	~24.5
Dispersion preparation	~0.1	Aging	~12.0
Crushed ice casting (Two step)	~0.5	Modification-replacement	~36.0
Freeze drying	~48.0	Drying	~24.0 (Estimate)
Total	~72.6	Total	~96.5

*We have ignored the preparation time of excipients.

** We have ignored the preparation time of fiber felts.

Revise details:

- Page 12, Line 26, added, “The whole process time from raw materials to products is also much lower than the traditional fiber felt/silica aerogel composites (Supplementary Table 1)¹³.”.

Comment 5 and 6: Fig. 4d presents a digital image showing an ultralight feature of ASNAs with an ultralow density of 0.59 mg/cm^3 . Except this image, however, all other properties and structures shown are those with a much higher density of 5 mg/cm^3 , as stated in Method (p.14 Line 322-323). This information should be included in the main text, instead of in Methods, to avoid any misunderstanding.

Associated with comment 5: The ultralight weight is an important consideration for aviation applications as proposed. As such, the authors should justify the use of ASNAs with 5 mg/cm^3 for studies of the properties and structures. The optimal concentration of $\text{Al}_2\text{O}_3\cdot\text{SiO}_2$ nanofibers and the final density of aerogels should also be discussed in detail.

Response: Thank you very much for your important comments and suggestions. We have added the information in the main text to avoid any misunderstanding.

Revise details:

- Page 7, Line 25, we added “To investigate mechanical property of the obtained ASNF aerogels, we first systematically investigated the effect of density regulation and size enlargement on the mechanical properties of the resultant materials. The densities of the ASNF aerogels can be readily regulated by changing the concentrations of the precursor dispersions, and the minimum density achieved is $0.59 \text{ mg}\cdot\text{cm}^{-3}$ (Supplementary Fig. 7). However, when the fiber concentration is too low (less than 0.1%), the cell wall consisted of the nanofibers will be loose during the subsequent freezing process, and it is even difficult to connect each other. When the fiber concentration is too high (higher than 1.8%), the nanofibers will be agglomerated and difficult to evenly disperse in the solution, and the obtained cell wall will be highly compact. Moreover, the thermal conductivity and elastic modulus of the obtained nanofiber aerogel increase as the density increases (Supplementary Table 1). Eventually, to achieve both low thermal conductivity and high elastic modulus, a density of 5 mg cm^{-3} was selected for test samples.”.

Supplementary Fig. 7. Density regulation of ASNF aerogels. **a** Density of aerogels *versus* concentration of nanofiber dispersion. **b** Weight measurement of ASNF aerogels. The sample with a volume of 18.0 cm⁻³ has a mass of 10.7 mg, which corresponds to a bulk density of 0.59 mg cm⁻³. **c-h** Compressive stress-strain curves of ASNF aerogels with the densities of 1.0, 2.5, 5.0, 12.0, and 16.0 mg cm⁻³.

Supplementary Table 2. Thermal conductivity and elastic modulus of the nanofiber aerogel with different bulk density.

Density (mg cm ⁻³)	Thermal conductivity (mW m ⁻¹ K ⁻¹)	Young's modulus (kPa)
0.6	24.01	0.25
1.0	24.38	0.50
2.5	25.00	0.98
5.0	26.00	3.95
12.0	28.28	8.17
16.0	29.21	10.36

Comment 7: Fig. 4f, only one thermal conductivity is recorded for each anisotropic aerogel (prepared from the SBS and directional freeze-casting processes). Report the thermal conductivities in both the alignment and transverse directions for anisotropic aerogels.

Response: Thank the reviewer for the insight comments. For lamellar sponges obtained by SBS¹⁴, anisotropic aerogels from directional freezing, the thermal conductivities along the axial and transverse direction have significant difference. As a result, the heat conduction along the hollow channels is usually two times higher the thermal conductivity perpendicular to the hollow channel. As the reviewer noted, we have added the test of the thermal conductivities in both the alignment and transverse directions for lamellar sponges and anisotropic aerogels in the revised manuscript.

Revise details:

- Page 12, Line 26, we added, “We also compared the thermal conductivity of our ASNf aerogels with those of the lamellar sponges and anisotropic aerogels along the axial and transverse direction (Fig. 4g). The sponges have a relatively high bulk density (~20 mg cm⁻³), corresponding to comparably higher thermal conductivities in both the transverse and axial directions than other two aerogels. The thermal conductivities of anisotropic ASNf aerogels are measured to be about 26 and 35 mW/m·K along the transverse and axial direction, respectively. Particularly, the thermal conductivity value in the transverse direction is comparable to that of the reported thermally insulating

silica nanofiber aerogels¹⁰, demonstrating the advantage of nanosized building blocks with large amounts of phonon barriers¹⁵. However, the aligned channels serve as highways for gas convection, while the aligned tube walls serve as channels for solid conduction in axial directions, which would increase the axial heat transfer.”

Fig. 4g. Thermal conductivity *versus* temperature profiles of lamellar sponges obtained by SBS, anisotropic aerogels from directional freezing, and our aerogels.

Comment 8: According to SI Section 6 and Methods, apart from the fabrication method, the composition and thermal treatment of aerogels are not identical. Therefore, the validity of thermal conductivity comparison between different fabrication methods (Fig. 4f) is questionable. The difference in composition between different samples should be clearly declared. In addition, the use of thermal treatments for directionally freeze-cast aerogels and ASNAs while no treatment for lamellar sponges should be fully justified.

Response: Thank you very much for your valuable comments and suggestions. I am sorry for the mistakes on the composition and thermal treatment of anisotropic ASNF aerogels. For the samples applied for thermal conductivity comparison, ASNFs with same and composition were used to ensure that the comparison is scientific. Moreover, the same nanofiber dispersion was used for directional freezing and

crushed ice manufacturing. Moreover, we are convinced that the drying and calcination conditions are also completely consistent. We also repeated this experimental process to ensure the reliability of the results. Thank you again for your valuable comments, which make our research more convincing.

Revise details:

- In Supplementary Materials, Page S9, Line 16, revised as “Typically, the used ASNFs nanofiber dispersion had identical composition with that for crushed ice manufacturing.”.
- In Supplementary Materials, Page S9, Line 19, revised as “Next, the precursors were calcined at 600 °C for 30 min to prepare the anisotropic ASNF aerogels with a bulk density of 5.0 mg cm⁻³.”

Comment 9: In Page 10 Line 214, the unit for bulk density should be mg cm⁻³.

Response: Thank you very much for your valuable comments and suggestions. We have revised it.

Revise details:

- Page 12, Line 26, the “~20 mg cm⁻¹” has revised as “~20 mg cm⁻³”.

Responses to Referee #2

The authors expose in this manuscript a variation of ice-templating enabling the production of large (m² range) pieces, with isotropic porous microstructures. The applicability of the process is then demonstrated with different materials (proof of concept).

The novelty of the process is its division into two freezing steps:

- A first step where only small bits of suspension (or solution) are ice-templated and scrapped from the cold surface repeatedly. This ensures the relatively rapid production of a large volume of frozen bits (which the authors refer to as “snow”).
- A second step where the “snow” is mixed with additional native suspension (to fill the gaps between the snow bits), and frozen, before being freeze-dried.

Since the novelty is in the process itself, I would have expected a deep, parametric investigation of the process and the influence of the various parameters, that would have provided a good overview and understanding of the process, its possibilities, control, and limitations. Instead, only a small part of the paper (approx. 20%) is devoted to the process, and almost no information and data are provided about its control and properties. The majority of the paper is about the proof-of-concepts materials obtained with this process. Although it's nice to demonstrate what the process can do, this poor balance makes the paper confusing and a bit shallow. Examples of these points are discussed below.

Response summary: Thank you for your constructive, detailed and accurate comments and suggestions. We also appreciate for your interest in the whole article including reading the details, understanding our ideas, and suggesting to the manuscript. In the past three months, we have carefully gone through and discussed these thoughtful and meaningful comments with our colleagues. We have gained a lot including the preciseness of writing, scientific depth, material performances, and experimental details. We have revised the manuscript according to the reviewer's suggestions. Moreover, we have carefully addressed, point by point, all the comments the reviewer has raised. We hope that the reviewer will find our responses satisfactory and convincing.

As suggested by the reviewer, the structure of the paper has been readjusted and we

have increased the content proportion of crushed ice casting method and formation mechanism. We have revised the manuscript according to the reviewer's suggestions and added the relevant experimental results mentioned by the reviewer. For example, we added the effect of roller speed on output and the ratio of ice to integrating agent on freezing time and the final structure. We also added the section headings to enhance the readability. A logical thinking chain of "preparation-mechanism-performance-application" has been formed.

Revise details:

- In Page 4, Line 1, we added, "The rotating speed was set as 50 rad min^{-1} , owing to a higher speed will cause the incomplete freezing of the nanofiber dispersion."
- Page 4, Line 5, we added, "The ratio of ice to integrating agent (nanofiber dispersion) can significantly affect the subsequent freezing time and the final structure. The excessive dosage of the dispersion causes the lamination of crushed ice and integrating agent, resulting in long freezing time and uneven structure. Meanwhile, the insufficient dosage makes it difficult to completely fill the gaps between the crushed ice, leading to considerable structural defects. According to experimental research, a 5:1 mass ratio for ice crystal and dispersion is appropriate for the following blending and freezing."
- Page 5, Line 22, we added, "The total required freezing time is 1.38 times shorter than that of conventional directional freezing process, and 1.92 times shorter than that of nondirectional freezing process, when producing 1.2 kg ice block (Fig. 1e and Supplementary Fig. 5). The whole process time from raw materials to products is also much lower than the traditional fiber felt/silica aerogel composites (Supplementary Table 1)¹³."
- Page 17, Line 6, we added, "The solution was rapidly transformed into nanofiber crushed ice through the crushed-ice-making machine at a rotating speed of 50 rad min^{-1} . The cryogenic drum has a diameter of 13 cm and a width of 11 cm with a surface temperature of $-20 \text{ }^{\circ}\text{C}$. Then, the obtained crushed ice was mixed with nanofiber suspensions (normal temperature) in a mass ratio of 5:1 to eliminate the gap between crushed ice. For typical experiments, a 200 g salad-like ice-dispersion mixture can be obtained after a

half-minute blending using a thermally insulated mixing tank and customized agitator paddles (Supplementary Fig. 24). Next, they were transferred to the desired mold, frozen in a low-temperature chamber ($-20\text{ }^{\circ}\text{C}$), and then freeze-dried for 18 h to obtain the ASNF aerogel precursors. Owing to the salad-like ice-dispersion mixture having a certain fluidity, it can fill the mold without leaving pores.”

- Page 3, Line 29, we added section headings, “Fabrication of isotropic nanofiber aerogels.”
- Page 6, Line 1, we added section headings, “Observation of the freezing process.”
- Page 7, Line 25, we added section headings, “Temperature-invariant mechanical performances.”
- Page 10, Line 15, we added section headings, “Mechanical analysis and simulations.”
- Page 12, Line 4, we added section headings, “Thermal Insulation performance.”
- Page 13, Line 26, we added section headings, “Proof-of-concept experiments.”

Comment 1: The paper is also filled with exaggerated and hype statements which I found annoying (“excellent”, “outstanding”, etc.) and not very scientific. Great science does not need hype.

Response: We are thankful for your constructive comments and suggestions. We have double-checked the manuscript and deleted all the exaggerated and hype statements (such as “excellent”, “outstanding”, “unique”, and “remarkably”) in the revised manuscript to ensure that all the claims revised on properties and performance are scientific and objective.

Revise details:

- Page 2, Line 1, “excellent” has been revised as “good”.
- Page 2, Line 1, “outstanding” has been deleted.
- Page 2, Line 28, “unique” has been deleted.

- Page 2, Line 19, “outstanding” has been revised as “high”.
- Page 9, Line 26, “remarkably” has been deleted.
- Page 12, Line 3, “excellent” has been revised as “high”.
- Page 14, Line 8, “excellent” has been deleted.
- Page 16, Line 24, “excellent” has been revised as “high”.

Comment 2: the term “snow” is fancy but misleading. Snow crystals are grown in very different conditions with a different physics (solid/vapor equilibrium). I think the authors should use a different term.

Response: Thank you for your important comments and suggestions. We agree that the term “snow” is inappropriate to describe the product in the RIC process. During the RIC process, the nanofiber dispersion was rapidly frozen on the surface of a rotating cryogenic drum to form ice with large area and scraped into crushed ice with small sized by the blade on the machine. We have changed the word into “crushed ice” in the revised manuscript.

Revise details:

- All “snow” or “snowflakes” has been revised as “crushed ice”.

Comment 3: Line 59-60. It is challenging to scale up the thick bulk material fabrication while maintaining consistent properties. This challenge is supposedly solved here but actually not demonstrated. No mention is made of the maximum thickness obtained, for example. From the figure 1b and f, it seems that the thickness is of ~7-8cm (?), which is about twice larger than what has been demonstrated by conventional freeze casting (pieces several cm thick has already been reported). What is the maximum thickness achieved here? Scaling up the lateral dimensions is not so challenging (several papers have already been published on this), the challenge in scale-up so far has always been the increase in thickness, which is not clearly reported and discussed here. From what is exposed here, I suspect it is actually not so simple, since the “snow” is mixed with additional suspension into a block that needs to be frozen completely, as in conventional freeze-casting. These limits are not discussed in the manuscript.

Response: Thank you for your meaningful comments and suggestions. We are very

sorry for bad expression on the progress in the improvement of ice crystal growth mechanism in thickness direction using crushed ice casting. We agreed that the challenge in scale-up so far has been the increase in thickness. Most surface growth processes would gradually slow down or even stop, owing to the gradual vanish of concentration or temperature gradient and reduction of mass or heat transfer. We must emphasize that increasing thickness is not the core of difficulty, but maintaining freezing efficiency. For conventional freezing methods, the greater the thickness, the lower the heat transfer efficiency, and the higher the energy and time costs involved in manufacturing materials.

- (i) Firstly, we compared the ice block production rate between the conventional nondirectional and directional freezing with crushed ice casting. Compared with the traditional ice-template method, our crushed ice casting method The total required freezing time is 1.38 times shorter than that of conventional directional freezing process, and 1.92 times shorter than that of nondirectional freezing process, when producing 1.2 kg ice block.
- (ii) We achieved the decoupling of the ice nucleation and growth processes. The 3D expansion of the growth sites will greatly accelerate the crystallization process, outperformance the 2D nucleation from directional freezing, especially in large-scale manufacturing. Moreover, we confirmed the high energy transfer efficiency of rotating freezing in the preparation of crushed ice through computational fluid dynamics (CFD) simulations¹⁶. The principle of heat and mass transfer enhancement in this thin-film rotating reactor has been widely reported¹⁷⁻²⁰.
- (iii) To probe the mechanism differences between directional freezing and crushed ice casting, we *in-situ* observed the cooling process under an optical-fluorescence microscope. We recorded the temperature profile during cooling, the slurry temperature decreased linearly before ice nucleation and a sudden temperature rise can be observed when the slurry temperature reached 1.3 °C, which is attributed to the heat release upon nucleation (Fig. 2b). After 24 seconds, the cooling slope is almost identical

with initial state, indicating the crystallization process has been completed. However, in the case of crushed ice casting strategy, all pre-existing and 3D random distributed ice crystals can act as the primitive growth sites. The adjacent crystals will unconstrainedly grow with no favorable location and orientations, and ultimately form a multidomain bulk materials. The ultrafast crystallization process (1.2 seconds) obtained from sequential optical images further confirmed this phenomenon.

- (iv) According to classical crystal theory, the transformation of liquid into solid at low temperature is divided into two parts: Nucleation requires a certain degree of undercooling and solidification environment; and the growth rate is proportional to the number of nucleation. On the basis of the above theory and observations, we propose here a possible mechanism for enhancing the cooling efficiency and controlling the 3D structures by crushed ice casting. We successfully combine these two mechanisms to achieve isotropic architectures in ice-templating materials. (i) Ice crystals rapid nucleation on rotating cryogenic drum can effectively increase heat transfer efficiency. (ii) Multi-point growth from three-dimensional distribution of crystal nucleus can effectively accelerate the formation of bulk crystals, resulting in a macroscopically cellular distribution of nanofibers.
- (v) Through blending the crushed ice and the dispersion evenly, the ice crystals are distributed uniformly and randomly after the mixture is poured into the pre-fabricated mold, and grow almost simultaneously in the freezing environment, thus ensuring the consistency of the structure as the size increases.

Revise details:

- Abstract, in Page 1, Line 29, we added, “Herein, we describe a general ice-templating technology through rotating freezing, crushing, and re-casting nanofiber slurry based on decoupling the ice nucleation and growth processes.”
- In Page 2, Line 26, we deleted, “It is challenging to scale up the thick bulk

material fabrication while maintaining consistent properties matching the variable application scenario.”

- In Page 5, Line 19, we added, “Compared with the traditional ice-template method, our crushed ice casting method decouple the ice nucleation and growth processes. The 3D expansion of the growth sites will greatly accelerate the crystallization process, outperformance the 2D nucleation from directional freezing, especially in thick sample manufacturing. The total required freezing time is 1.38 times shorter than that of conventional directional freezing process, and 1.92 times shorter than that of nondirectional freezing process, when producing 1.2 kg ice block.”
- In Page 5, Line 19, we added, “Large-scale samples often encounter issues of uniformity and mechanical property degradation. In this study, the large-scale samples (35.0×34.0×2.5 cm³) exhibited similar compression-recovery curves at different parts, which is almost identical to with the small size samples tested in the following (Supplementary Fig. 8). These observations demonstrated that crushed ice casting processes can be extremely powerful for nanofiber aerogel formation because these materials could be easily scaled without alteration of the mechanical properties of the resultant materials.”

Supplementary Fig. 8. Performance of different parts at a large-scale sample. a In-plane optical images of large-scale sample. b Images of ASNF aerogels on thickness direction. c Optical images showing the sampling point. c Compressive stress-strain curves of ASNF aerogels at different parts from A to E. d Micromorphology and thermal conductivity of ASNF aerogels at different parts from A to E.

Comment 4: “these building blocks often results in interconnected macro-scale open pores, whose dimensions are greater than the standard free path of air”. The authors imply here (and later in the paper) that their approach is better in this regard. However:

- no pore characterization and pore size distribution were reported here
- the SEM pictures reveal that the porous structure is clearly macroporous (micro- or mesopores could also be present but not visible on the SEM pics) and interconnected.

I thus don't see anything unique or specific in these structures. Such strong claims should be supported by experimental data, which is not the case here.

Response: Thank you for your constructive comments and suggestions. I am sorry the demonstration of our pore structure is unreasonable for reading. As the reviewer noted, we have carried out more detailed characterization of the pore structure of the isotropic ASNF aerogels. The size of pores in isotropic ASNF aerogels are contributed to the ice crystals and the compositions. According to IUPAC classification, the pores are classified in to the following groups according to their diameter: those with size less than 2 nm are called micropores; the size larger than 50 nm is called macropore; those between 2 nm and 50 nm are called mesopores. Owing the nanofibers are assembled on the surface of the ice crystal, the final aperture size of the aerogel will depend on the size of the ice crystal. As shown in Fig 4i, the average size of the ice blocks are 40.86 μm ,

As suggested, we tested the area pore size distribution of isotropic ASNF aerogels by an automated mercury porosimeter. This is a standard test method for pore size distribution of porous materials. The obtained average pore size is 39.86 μm , and the pore size is mainly distributed in the range of 6 to 106 μm , which is corresponded well with the ice block size. In additional, the nanofiber cell walls of ASNF aerogels consist of numerous mesopores due to the adding of commercial silica aerogel powders, which exhibited a small pore size of 2-4 nm, as shown in Supplementary Fig. 18, characterized by the nitrogen sorption isotherms test. These results indicate the ASNF aerogels are mainly consisted of macropores, while some mesoporous materials are embedded in the structure.

Moreover, as for SEM images, we can clearly observe the interconnected macroporous cellular structure, bonded nanofibers, and mesoporous silica aerogel powders (Fig.1d, 4a, and Supplementary Fig. 11.).

Revise details:

- In Page 1, Line 29, “These highly porous skeletons could effectively reduce solid heat conduction. However, these building blocks often result in interconnected macro-scale open pores, whose dimensions are greater than the standard mean free path of air (69 nm); and fail to restrict the free movement of molecules and mitigate the gaseous heat conduction.” have been replaced by

“However, because of the channel dimensions are much greater than the standard mean free path of air (69 nm), which fail to restrict the free movement of molecules and mitigate the gaseous heat conduction²¹.”

- In Page 9, Line 21, we added, “The superelastic characteristics and negative Poisson's ratios are mainly derived from the bonded nanofiber networks and interconnected separate cellular structures (Supplementary Fig. 11).”
- In Page 12, Line 18, we added, “The macropore size distribution of ASNF aerogels was characterized by an automated mercury porosimeter (Fig. 4f). The average size of macropore in the ASNF aerogels was 39.86 μm and the porosity was estimated as 90.73%.”
- In Page 12, Line 23, we added, “Contrasting the nitrogen sorption isotherms of these materials to our aerogels showed that our aerogels had the highest Brunauer-Emmet-Teller (BET) specific surface area ($21.89 \text{ m}^2 \text{ g}^{-1}$) and Barrett-Joyner-Halenda (BJH) average pore sizes (3.86 nm), owing to the existence of commercial silica aerogel powders (Supplementary Fig. 18 and Table 3).”
- We added Supplementary Fig. 11.

Figure 4f. Pore size distribution of ASNF aerogels derived from automated mercury porosimeter.

Supplementary Fig. 11. Morphology of ASNf aerogels. **a** Initial nanofibers are randomly stacked without interconnected points. **b** Microstructures of ASNf aerogels at different magnifications demonstrating the hierarchical nanofibrous cellular architecture. The aerogels were macroscopically interconnected through nanofiber interweaving, and microscopically bonded by porous inorganic binders and silica aerogels after 3D reconstruction.

Supplementary Fig. 18. Brunauer-Emmett-Teller analysis. **a** N₂ sorption isotherms of different materials. **b** Pore size distribution derived from Barrett–Joyner–Halenda analysis.

Supplementary Table 3. The pore structural parameters of the samples

Samples	BET Specific surface area ($\text{m}^2 \text{g}^{-1}$)	BJH Pore volume ($\text{cm}^3 \text{g}^{-1}$)	Average pore diameter (nm)
Lamellar sponges	1.54	0.0116	2.19
Anisotropic aerogels	16.15	0.0359	3.28
Isotropic aerogels	21.89	0.0444	3.86

Comment 5: the “snow” is mixed with additional suspension and cast into blocks that need to be frozen completely. Is the additional suspension cooled before being mixed with the snow? Is the mix homogenized? Are there any precautions to take to avoid thawing? How much pressure is applied to packed the mix and ensure no pores are left? No mention is made of how long it takes to freeze such large blocks. No mention is made either about the required freeze-drying time. Too little information about this important step is provided in the manuscript.

Response: Thank you for your important comments and suggestion. As the reviewer suggested, more information about the crushed ice casting method have been provided in the revised manuscript.

- (1) In the fabrication process, the additional suspension was not cooled before being blending with the crushed ice. Normal temperature dispersion is sufficient because the mass ratio of dispersion to crushed ice is 1:5.
- (2) For typical experiments, a 200 g salad-like ice-dispersion mixture can be obtained after half-minute blending using a thermally insulated mixing tank and customized agitator paddles. The time spent in this process can be ignored for the whole process.
- (3) Supplementary Fig. 24 show the optical images of crushed-ice-dispersion mixture. In the blending process, the ice will be further refined and melting under the action of external force, ensuring the uniform distribution of the ice and water. Partially melting of ice crystallization are inevitable during the mixing of ice and solution, although we use thermally insulated containers. The homogeneity of the mixture is confirmed by the stability of the mechanical properties of different parts at a large-scale sample.
- (4) Owing to the salad-like ice-dispersion mixture has a certain fluidity, it can fill the mold without leaving pores.

- (5) As shown in Fig 1e, the time required for freezing is related to the mass of the sample. The total required freezing time of crushed ice casting (two step) is 1.38 times shorter than that of conventional directional freezing process, and 1.92 times shorter than that of nondirectional freezing process, when producing 1.2 kg ice block (Fig. 1e and Supplementary Fig. 5).
- (6) We added the freeze-drying time in Supplementary Table 1. We compared the whole process time from raw materials to products between isotropic ASNF aerogels and the traditional fiber felt/silica aerogel composites (Use $300 \times 300 \times 10 \text{ mm}^3$ sample as standard). The whole process time from raw materials to products is also much lower than the traditional fiber felt/silica aerogel composites (Supplementary Table 1)¹³.

Revise details:

- In Page 5, Line 21, We added, “The total required freezing time is 1.38 times shorter than that of conventional directional freezing process, and 1.92 times shorter than that of nondirectional freezing process, when producing 1.2 kg ice block (Fig. 1e and Supplementary Fig. 5). The whole process time from raw materials to products is also much lower than the traditional fiber felt/silica aerogel composites (Supplementary Table 1)¹³.”
- In Page 17, Line 7, we added, “The obtained crushed ice was mixed with nanofiber dispersions (normal temperature) in a mass ratio of 5:1 to eliminate the gap between crushed ice.”
- In Page 17, Line 9, we added, “For typical experiments, a 200 g salad-like ice-dispersion mixture can be obtained after half-minute blending using a thermally insulated mixing tank and customized agitator paddles (Supplementary Fig. 24). Next, they were transferred to the desired mold, frozen in a low-temperature chamber ($-20 \text{ }^\circ\text{C}$), and then freeze-dried for 18 h to obtain the ASNF aerogel precursors.”
- In Page 17, Line 13, we added, “Owing to the salad-like ice-dispersion mixture has a certain fluidity, it can fill the mold without leaving pores.”
- We added Supplementary Fig. 24. Optical images of samples and agitator paddles. a ASNF aerogels with diverse shapes. The ASNF aerogels with an

isotropic structure could be molded into arbitrary shapes to fit the needs of the current diversified battery or module forms. **b** Customized agitator paddles for mixing crushed ice and nanofiber dispersion. **c** Optical images of crushed-ice-dispersion mixture.

- We added, “Supplementary Table 1. Whole process time from raw materials to products comparison between isotropic ASNF aerogels and the traditional fiber felt/silica aerogel composites (Use $300 \times 300 \times 10 \text{ mm}^3$ sample as standard).”

Supplementary Fig. 24. Optical images of samples and agitator paddles. a ASNF aerogels with diverse shapes. The ASNF aerogels with an isotropic structure could be molded into arbitrary shapes to fit the needs of the current diversified battery or module forms. **b** Customized agitator paddles for mixing crushed ice and nanofiber dispersion. **c** Optical images of crushed-ice-dispersion mixture.

Fig. R1. (Left) Comparison of ice block production rate between the conventional directional and nondirectional freezing with crushed ice casting. **(Right)** Production rate of crushed ice manufacturing and crushed ice re-casting steps.

Supplementary Table 1. Whole process time from raw materials to products comparison between isotropic ASNF aerogels and the traditional fiber felt/silica aerogel composites (Use $300 \times 300 \times 10 \text{ mm}^3$ sample as standard).

Astropic ASNF aerogels*		Fiber felt/silica aerogel composites**	
Procedures	Time (h)	Procedures	Time (h)
Nanofiber preparation	~24.0	Gel preparation	~24.5
Dispersion preparation	~0.1	Aging	~12.0
Crushed ice casting (Two step)	~0.5	Modification-replacement	~36.0
Freeze drying	~48.0	Drying	~24.0 (Estimate)
Total	~72.6	Total	~96.5

*We have ignored the preparation time of excipients.

** We have ignored the preparation time of fiber felts.

Comment 6: “the SC process was easily scaled in a linear fashion to over 1.2 m² area without alteration”. Where is the data supporting this statement? I only see a (nice) picture of a very large piece, but I don’t see a comparison of the microstructure/properties for pieces of increasingly large dimensions.

Response: Thank the reviewer for the valuable comments. With respect to the production of large dimensions, the fabrication process was consistent with the process for pieces of small dimensions, including spinning, homogenizer, freezing into crushed ice, freezing dryer and calcination. Therefore, our preparing method of the nanofiber aerogel samples is highly scalable (Fig. R1, these images have been

included throughout the article).

In the case of fabrication process for large-scale aerogels, there are nothing changed except the enlargement capacity of the freeze-dryer. Thus, the process still followed the forming mechanism of crushed-ice-template process. As shown in Supplementary Fig. 8, we have produced ASNAs with size of $35.0 \times 34.0 \times 2.5 \text{ cm}^3$. Then five positions were randomly selected on the large size samples to characterize the morphology, mechanical properties and thermal properties, thus confirming that the method is applicable to the preparation of large size samples and will not affect the structure and properties.

Fig. R2. ASNF aerogel samples with different sizes and shapes

Supplementary Fig. 8. Performance of different parts at a large-scale sample. **a** In-plane optical images of large-scale sample. **b** Images of ASNF aerogels on thickness direction. **c** Optical images showing the sampling point. **c** Compressive stress-strain curves of ASNF aerogels at different parts from A to E. **d** Micromorphology and thermal conductivity of ASNF aerogels at different parts from A to E.

Revise details:

- In Page 5, Line 8, We removed, “without alteration of the mechanical properties of the resultant materials.”
- In Page 9, Line 3, We added, “Large-scale samples often encounter issues of uniformity and mechanical property degradation. In this study, the large-scale samples (35.0×34.0×2.5 cm³) exhibited similar compression-recovery curves, thermal conductivity, and micromorphology at different parts (Supplementary Fig. 8), which is almost identical to with the small size samples tested in the

following. These observations demonstrated that crushed ice casting processes can be extremely powerful for nanofiber aerogel formation because these materials could be easily scaled without alteration of the mechanical and thermal insulating properties of the resultant materials.”

Comment 7: Many claims are made about the supposed advantages of the process, but these advantages are not really discussed, and no mention is made of the limits. For example, the SC process is more complex (more steps) than the standard freeze-casting process. The first freezing step may be faster, but a second freezing stage is required. Overall, it is very likely that the overall complexity and thus cost of the process is at least similar if not greater than conventional freeze-casting.

Response: Thank you for your constructive comments and suggestions. Indeed, we claim too many advantages, but insufficient evidence and disadvantages in the previous version. We have added discussion the advantages and disadvantages of the crushed ice casting method to fabricate the aerogels in the revised manuscript.

(1) In contrast with the traditional lamellar aerogels formed by directional freezing, our method decouples the nucleation and growth processes, thus, the preparation process is relatively complex. However, the directional freezing process usually encumbered by the great nucleation resistance and low heat transfer capacity owing to the relatively low thermal conductivity of ice. In the crushed ice casting process, the 3D expansion of the growth sites will greatly accelerate the crystallization process, outperformance the 2D nucleation from directional freezing, especially in large-scale manufacturing. Moreover, we confirmed the high energy transfer efficiency of rotating freezing in the preparation of crushed ice through computational fluid dynamics (CFD) simulations¹⁶. The principle of heat and mass transfer enhancement in this thin-film rotating reactor has been widely reported¹⁷⁻²⁰. Therefore, the total required freezing time is 1.38 times shorter than that of conventional directional freezing process, and 1.92 times shorter than that of nondirectional freezing process, when producing 1.2 kg ice block (Fig. 1e and Supplementary Fig. 5).

(2) Past attempts to modulate thermal insulation properties focused on developing a family of aerogels using directional freezing method based on

designs of flexible building blocks with ultralight features. The heat conduction across hollow channels (transverse heat transfer) of the samples could be effectively reduced with an extremely low thermal conductivity of $0.02 \text{ W m}^{-1} \text{ K}^{-1}$ owing to the highly tortuous heat transfer path and inhibited air diffusion. However, because the channel dimensions are much greater than the standard mean free path of air (69 nm), which fails to restrict the free movement of molecules and mitigate the gaseous heat conduction²¹. Meanwhile, the aligned channel wall is also beneficial to solid heat conduction. As a result, the heat conduction along the hollow channels (axial heat transfer) is usually two times higher than the transverse thermal conductivity. In the crushed ice casting process, the nanofibers are assembled into interconnected macropores and isotropic structure in three-dimensional space under the induction of crushed ice crystals. Moreover, these macroporous separate cellular structures composed by nanofibers built tortuous and complexity solid conduction paths, slowing down the heat transfer efficiency (Fig. 4h). At the nanoscale, the abundant number of mesopores could restrain the free movement of molecules and significantly decrease the gaseous heat conduction (Knudsen effect)²¹, resulting in increased interfacial thermal resistance.

Fig. R1. **(Left)** Comparison of ice block production rate between the conventional undirectional and undirectional freezing and crushed ice casting. **(Right)** Production rate of crushed ice manufacturing and crushed ice re-casting steps.

Revise details:

- Page 5, Line 20, added “The total required freezing time is 1.38 times shorter

than that of conventional directional freezing process, and 1.92 times shorter than that of nondirectional freezing process, when producing 1.2 kg ice block (Fig. 1e and Supplementary Fig. 5).”

- Page 17, Line 9, added in Methods, “When comparing the ice block production rate of the conventional unidirectional and directional freezing to crushed ice casting, the same mold size (18×18 cm² in-plane) and freezing environment (−20 °C) were adopted.”

Comment 8: “This was noteworthy as SC technology was incompatible with slurry-forming technology”. It doesn’t understand this statement, could you please elaborate?

Response: Thank you for the constructive comment. We are sorry for the careless mistake. SC technology was compatible with slurry-forming technology. Almost all fibers or powders that can be made into slurry can be turned into crushed ice by this method. Here, a general, controllable ice-templating strategy has been developed to large-scale fabricate a kind of isotropic aerogels composed of spatially well-defined nanofiber assemblies (e.g. aluminosilicate nanofibers (ASNFs), polyacrylonitrile nanofibers (PNFs), *p*-aramid nanofibers (ANFs), cellulose nanofibers (CNFs), and graphite nanotubes (GNTs)). We have corrected the mistake.

Revise details:

- In Page 5, Line 25, We added, “Here, a general, controllable ice-templating strategy has been developed to large-scale fabricate a kind of isotropic aerogels composed of spatially well-defined nanofiber assemblies (e.g. aluminosilicate nanofibers (ASNFs), polyacrylonitrile nanofibers (PNFs), *p*-aramid nanofibers (ANFs), cellulose nanofibers (CNFs), and carbon nanotubes (CNTs).”
- In Page 5, Line 25, We removed, “This was noteworthy as SC technology was incompatible with slurry-forming technology.”

Comment 9: “In a snow slurry system, the ice crystal tends to nucleate simultaneously”. If I understood correctly the process, the “snow” scrapped from the rotating drum is collecting and packed. The ice crystals are thus already present in the snow and not thawed at this stage. The ice crystals in the snow thus continue growing, there is no need to nucleate novel crystals.

Response: Thank you very much for your constructive comments and suggestions. We are sorry for the careless mistake. We agree that the ice crystals continue growing rather than nucleating. Therefore, the freeze process will be faster due to the omission of nucleation time. As noted, we have revised the relevant statements. As mentioned in the manuscript, when freezing slurry with the same mass, freezing the mixture of crushed ice and slurry will significantly increase the freezing rate, thus improving the efficiency of sample growth and preparation.

Revise details:

- In Page 7, Line 14, we added, “However, in the case of crushed ice casting strategy, all pre-existing and 3D random distributed ice crystals can act as the primitive growth sites.”
- In Page 7, Line 16, we added, “Owing to no nucleation process exist in the system, the setting temperature and slurry temperature are basically consistent (Fig. 2g). Moreover, the ultrafast crystallization process (1.2 seconds) obtained from sequential optical images further confirmed this phenomenon (Fig. 2h, i, and Supplementary Video 5).”

Fig. 2. Proposed freezing mechanism. a Schematic illustration of the directional freezing process showing successive nucleation and preferential growth. b Temperature variation over time during directional freezing. c-d In situ observation on directional freezing using an optical-fluorescence microscope. e Schematic illustration of setup for crushed ice casting observation. f Schematic illustration of the crushed ice casting process showing ice crystals grow around the existing crystals with different locations and grow randomly into various orientations, finally forming a multidomain pattern. g Temperature variation over time during crushed ice casting. h–i In situ observation on crushed ice casting using a optical-fluorescence microscope.

Comment 10: “We successfully realize complex architectures in snow-casted materials”. It’s not particularly complex, it’s either random (and somewhat isotropic), or already reported before (e.g., Fig S6). “This structure exhibited a special micro-orientation and macro-isotropic nature”. I don’t see what’s special (what does “macro-isotropic” means?) here, more explanations should be provided. The structure shown in Fig S6 is typical of freeze-casted fibrous structures (e.g. freeze-casted carbon nanotubes structures), which has been reported in numerous papers before. It’s nice but I don’t see anything special or novel here.

Response: Thank you very much for the insightful comments. We agreed that the structure is not particularly complex, and similar structure has been reported using foaming method (bubble templated)²²⁻²⁴, cellulose-based nondirectional freezing method²⁵, and freeze-casted carbon nanotubes structures^{26, 27}, but not common on ceramic nanofiber aerogels. Indeed, this expression is inappropriate in the manuscript. We agreed that the structure is not particularly special. Similar claim in the manuscript has been revised.

The “macro-isotropic” means that the macroscopic materials obtained by the crushed ice casting technology exhibited isotropic properties. As the reviewers suggested, such claims have been revised. The cellular structure of ASNF aerogels is consisted of cellular pores mainly ranging from 10 to 30 μm . The formation of the aerogels depends on the assembling of nanofibers on the moving solidification ice front. For the samples obtained from directional freezing, the lamellar ice crystals grow from the initial crystal nucleus formed near the cold plate. Thus, the growth direction would orient in an axial direction from cold bottom to hot top. However, the

pre-existed crystal nucleus was randomly distributed in the whole sample for the crushed ice casting method. Thus, the ice crystal grows up almost the same time to form the cellular structure which is isotropic in the macro-scale. Thus, an effective strategy to fabricate isotropic aerogels with the uniform cellular structure was reported in this manuscript.

More detailly, past attempts to modulate thermal insulation properties focused on developing a family of ceramic aerogels using directional freezing method based on designs of flexible building blocks with ultralight features. The heat conduction across hollow channels (transverse heat transfer) of the samples could be effectively reduced with an extremely low thermal conductivity of $0.02 \text{ W m}^{-1} \text{ K}^{-1}$ owing to the highly tortuous heat transfer path and inhibited air diffusion. However, because the channel dimensions are much greater than the standard mean free path of air (69 nm), which fails to restrict the free movement of molecules and mitigate the gaseous heat conduction²¹. Meanwhile, the aligned channel wall is also beneficial to solid heat conduction. As a result, the heat conduction along the hollow channels (axial heat transfer) is usually two times higher than the transverse thermal conductivity. In the crushed ice casting process, the nanofibers are assembled into interconnected macropores and isotropic structure in three-dimensional space under the induction of crushed ice crystals. Moreover, these macroporous separate cellular structures composed by nanofibers built tortuous and complexity solid conduction paths, slowing down the heat transfer efficiency (Fig. 4h). At the nanoscale, the abundant number of mesopores could restrain the free movement of molecules and significantly decrease the gaseous heat conduction (Knudsen effect), resulting in increased interfacial thermal resistance.

Revise details:

- In Page 7, Line 24, we added, “Past attempts to modulate thermal insulation properties focused on developing a family of aerogels using directional freezing method based on designs of flexible building blocks with ultralight features. The heat conduction across hollow channels (transverse heat transfer) of the samples could be effectively reduced with an extremely low thermal conductivity of $0.02 \text{ W m}^{-1} \text{ K}^{-1}$ owing to the highly tortuous heat transfer path and inhibited air diffusion. However, because the channel dimensions are

much greater than the standard mean free path of air (69 nm), which fails to restrict the free movement of molecules and mitigate the gaseous heat conduction²¹. Meanwhile, the aligned channel wall is also beneficial to solid heat conduction. As a result, the heat conduction along the hollow channels (axial heat transfer) is usually two times higher than the transverse thermal conductivity.”

- In Page 7, Line 24, we replaced “complex” by “isotropic”.
- In Page 9, Line 30, we removed, “This structure exhibited a special micro-orientation and macro-isotropic nature, resulting in isotropic mechanical and thermal properties.”

Comment 11: Fig 2D: I do not understand the figure. What is the direction of the temperature gradient or sample with respect to the surface?

Response: Thank you very much for your valuable comments. We deleted the previous version of Fig.2d, and readjusted the figures For Fig 2a, the temperature gradually increases from bottom to top. For Fig 2c, d, the temperature gradually increases from left to right. The bottom of the schematic figure is equivalent to the left end of the experiment, where the temperature is lowest. Thus, for sample fabricated by freezing-casting, the lamellar ice crystals grow from the initial crystal nucleus formed near the cold copper plate, resulting the growth direction would orient in an axial direction from cold bottom to top.

While for sample fabricated by crushed ice casting, the ice act as the initial crystal nucleus and grow randomly to form ice crystal with different shapes. The temperature is uniform. We have added the temperature gradient direction in the figures.

Revise details:

- In Page 6, Line 6, we replaced Fig 2 by a new Figure.
- In Page 7, Line 2, we added, “Typically, the ice crystals nucleate successively from the surface of the copper platform and grow preferentially in the vertical direction along the temperature gradient, resulting in a macroscopically long-range aligned lamellar distribution.”

Fig. 2. Proposed freezing mechanism. a Schematic illustration of the directional freezing process showing successive nucleation and preferential growth. b Temperature variation over time during directional freezing. c-d In situ observation on directional freezing using an optical-fluorescence microscope. e Schematic illustration of setup for crushed ice casting observation. f Schematic illustration of the crushed ice casting process showing ice crystals grow around the existing crystals with different locations and grow randomly into various orientations, finally forming a multidomain pattern. g Temperature variation over time during crushed ice casting. h-i In situ observation on crushed ice casting using an optical-fluorescence microscope.

Comment 12: Fig 3D: as far as I can tell, this not a standard test. It is thus not possible to compare the behavior with other materials.

Response: Thank you for your insightful comments. Indeed, this method is not a standard method for testing fatigue resistance. As shown in Fig 3d, we recorded the variations in maximum stress value and plastic deformation with the change of the

number of compression recovery cycles, thus characterizing the attenuation of strength or stiffness, which further verified the structural robustness. It is also a reported characterization method to verify the mechanical properties of aerogels, which was published on *Science* **310**, 1307-1310 (2005), *Science advances* **4**(4), eaas8925 (2018), *Angewandte Chemie* **132**(21): 8362-8369(2020), *Chem* **4**, 544-554 (2018)., *Advanced Functional Materials*, **30**(49), 2005928 (2020), *Science* **363**, 723-727 (2019), and *Nat Commun* **7**, 12920 (2016). The test result is not suitable for lateral comparison due to different compression levels, but can be used as an evidence for stable elasticity.

Revise details:

- All the “fatigue resistance” and been replaced by “damage tolerance”.

Comment 13: “The bulk density [...] could be arbitrarily adjusted in the range of 0.59 to 20 mg/cm³”. Where are the data? What’s the variability? How reproducible is the process?

Response: Thank you for your valuable comment and suggestions. As mentioned in the manuscript, the densities of the ASNF aerogels can be regulated by changing the concentrations of the precursor dispersions. This process is highly reproducible, which has also been confirmed in many other literature^{5, 10, 28-34}. We have measured the correlation between density and concentration, as shown in **Supplementary Fig. 7**. The concentrations of the precursor dispersion can range from 0.05% to 1.8%. For concentrations overtop 1.8%, the high concentration of nanofiber makes the agitation resistance great, which leads to the fiber is difficult to disperse evenly. For concentrations below 0.05%, too few fibers make it difficult to construct an effective cell wall, leaving the structure loose or barely formed. Therefore, the sample will lose its elastic properties and be prone to result collapse.

Revise details:

- Page 7, Line 25, we added, “To investigate mechanical property of the obtained ASNF aerogels, we first systematically investigated the effect of density regulation and size enlargement on the mechanical properties of the resultant materials. The densities of the ASNF aerogels can be readily regulated by changing the concentrations of the precursor dispersions, and

the minimum density achieved is $0.59 \text{ mg}\cdot\text{cm}^{-3}$ (Supplementary Fig. 7). However, when the fiber concentration is too low (less than 0.1%), the cell wall consisted of the nanofibers will be loose during the subsequent freezing process, and it is even difficult to connect each other. When the fiber concentration is too high (higher than 1.8%), the nanofiber will be agglomerated and difficult to evenly disperse in the solution, and the obtained cell wall will be highly compact. Moreover, the thermal conductivity and elastic modulus of the obtained nanofiber aerogel increase as the density increases (Supplementary Table 1). Eventually, to achieve both low thermal conductivity and high elastic modulus, a density of 5 mg cm^{-3} was selected for test samples.”.

Supplementary Fig. 7. Density regulation of ASNF aerogels. **a** Density of aerogels versus concentration of nanofiber dispersion. **b** Weight measurement of ASNF aerogels. The sample with a volume of 18.0 cm^{-3} has a mass of 10.7 mg , which corresponds to a bulk density of 0.59 mg cm^{-3} . **c-h** Compressive stress-strain curves

of ASNF aerogels with the densities of 1.0, 2.5, 5.0, 12.0, and 16.0 mg cm⁻³.

Supplementary Table 2. Thermal conductivity and elastic modulus of the nanofiber aerogel with different bulk density.

Density (mg cm ⁻³)	Thermal conductivity (mW m ⁻¹ K ⁻¹)	Young's modulus (kPa)
0.6	24.01	0.25
1.0	24.38	0.50
2.5	25.00	0.98
5.0	26.00	3.95
12.0	28.28	8.17
16.0	29.21	10.36

Comment 14: I am not confident to evaluate the modelling part of the specific materials properties reported here.

Response: CFD model has been used to simulate the heat transfer process in mature cases. The N-S constitutive equation is used for modeling, and the empirical parameters are less in the process, so the reliability is strong. In addition, the model can simulate the information that cannot be measured during the experiment, such as the temperature change of thin ice in the drum. And then guide the experimental process. The model can simulate the different working conditions of the experiment, and then select the appropriate process conditions. Therefore, the number of experiments can be greatly reduced, thus saving human, material and financial resources. Therefore, it is necessary to use the model to simulate the process.

Comment 15: Overall, the paper is not focused. Too little details are provided about the novel process and its control and limits, and too many information is provided about a myriad of very different materials. This variation of freeze-casting appears thus promising but too little information is provided about the process at this stage to make it a convincing, real advancement.

Response: Thank you for your constructive, detailed and accurate comments and suggestions. We also appreciate for your interest in the whole article including reading the details, understanding our ideas, and suggesting to the manuscript. In the past

three months, we have carefully gone through and discussed these thoughtful and meaningful comments with our colleagues. We have gained a lot including the preciseness of writing, scientific depth, material performances, and experimental details. We have addressed these comments point by point by performing additional theoretical and experimental studies. We believe that with your help, the quality and impact of the revised paper have been evaluated to a higher level. Although we have made a lot of efforts, there were still some errors and omissions in the manuscript. We hope you can continue to point out them.

Responses to Referee #3

The authors describe the methodology and successful synthesis of an ultralight, large-scale, thermal super insulating, and flexible nanofiber aerogel using a continuous rotating ice crystallization. The team also demonstrated the capability of a 5 mm thick film to decrease the risk of thermal propagation, which is a major challenge with lithium-ion batteries and an area of the field with a continuous need of advancements.

Comment 1: I recommend the authors to discuss the challenges and trade-offs of mitigating the risk of propagation and the added weight. What increase in parasitic mass and volume ratios do the nanofiber aerogel film introduce to the cell stack? Parasitic mass ratio is defined as mass of cell stack divided by mass of cells only. This may show that adding thermal capacitance to the cell stack is not an effective mitigation strategy for propagation resistance.

Response: Thank you very much for your comments and suggestions. As the market demands for electric vehicles and power grids, large multi-cell Li-ion batteries in vast quantities are required. However, for multi-cell lithium-ion batteries, the cell-to-cell propagation of thermal runaway generated by a single battery presents a high risk for battery users. Currently existing measures for preventing TR propagation are divided in to four different concepts, including, (i) the use of fire walls with very low thermal conductivity, (ii) high thermally conductive materials to provide quick heat dissipation, (iii) nature convection and forced cooling that circulate coolants around the cells, and (iv) phase-changing chemicals with non-reversible endothermic phase transitions.

Traditional fire walls will add weigh and volume significantly, thus diminishing gravimetric and volumetric energy densities of the battery. While for phase change materials, one of the main disadvantages is the flash point of the customarily employed kinds of paraffin, which contribute significantly to the fire load. In general, air cooling system is limited by the low thermal conductivity and small specific heat capacity of air. Although the use of electric fans will increase the airflow rate, it will lead to additional energy consumption costs. While for the liquid cooling system, the flow channels and cooling medium could indeed result in large space occupation and weight increasing. And the hoses, pipes, or mini channels will increase the complexity, which is costly and will occupy a large volume in the battery module and pack.

For ceramic fiber aerogels prepared by the crushed ice template method, we calculated the increase of parasitic mass and volume ratio after adding aerogels according to the comments of reviewers. For the sample with a thickness of 0.5 cm, the density of the sample is 5 mg cm^{-3} , which can be ignored compared with the mass of the battery pack. The ASNF aerogels could completely block the TR propagation without additional assistants and specific energy loss. The volumetric energy density of the module is determined as $\sim 677.6 \text{ Wh L}^{-1}$ in the initial state and $\sim 589.3 \text{ Wh kg}^{-1}$ with ASNF aerogels.

Revise details:

Page 14, Line 30, we added. “In stark contrast, our ASNF aerogels could completely block the TR propagation without additional assistants and specific energy loss. The volumetric energy density of the module is determined as $\sim 677.6 \text{ Wh L}^{-1}$ in the initial state and $\sim 589.3 \text{ Wh kg}^{-1}$ with ASNF aerogels.”

REFERENCES

1. Yu ZL, *et al.* Bioinspired polymeric woods. *Sci Adv* **4**, eaat7223 (2018).
2. Gao HL, *et al.* Super-elastic and fatigue resistant carbon material with lamellar multi-arch microstructure. *Nat Commun* **7**, 12920 (2016).
3. Qin B, *et al.* A Petrochemical-Free Route to Superelastic Hierarchical Cellulose Aerogel. *Angew Chem Int Ed* **n/a**, (2022).
4. Li C, *et al.* Temperature-Invariant Superelastic and Fatigue Resistant Carbon Nanofiber Aerogels. *Adv Mater*, e1904331 (2019).
5. Fu Q, *et al.* Highly Carboxylated, Cellular Structured, and Underwater Superelastic Nanofibrous Aerogels for Efficient Protein Separation. *Adv Funct Mater* **29**, (2019).
6. Li Y, Cao L, Yin X, Si Y, Yu J, Ding B. Semi-Interpenetrating Polymer Network Biomimetic Structure Enables Superelastic and Thermostable Nanofibrous Aerogels for Cascade Filtration of PM2.5. *Adv Funct Mater* **30**, (2020).
7. Wang F, Si Y, Yu J, Ding B. Tailoring Nanonets-Engineered Superflexible

- Nanofibrous Aerogels with Hierarchical Cage-Like Architecture Enables Renewable Antimicrobial Air Filtration. *Adv Funct Mater*, (2021).
8. Wang F, *et al.* In situ Synthesis of Biomimetic Silica Nanofibrous Aerogels with Temperature-Invariant Superelasticity over One Million Compressions. *Angew Chem, Int Ed Engl* **59**, 8285-8292 (2020).
 9. Si Y, Yu J, Tang X, Ge J, Ding B. Ultralight nanofibre-assembled cellular aerogels with superelasticity and multifunctionality. *Nat Commun* **5**, 5802 (2014).
 10. Si Y, Wang X, Dou L, Yu J, Ding B. Ultralight and fire-resistant ceramic nanofibrous aerogels with temperature-invariant superelasticity. *Sci Adv* **4**, eaas8925 (2018).
 11. Bai H, Chen Y, Delattre B, Tomsia AP, Ritchie RO. Bioinspired large-scale aligned porous materials assembled with dual temperature gradients. *Sci Adv* **1**, e1500849 (2015).
 12. Zhao N, Li M, Gong H, Bai H. Controlling ice formation on gradient wettability surface for high-performance bioinspired materials. *Sci Adv* **6**, eabb4712 (2020).
 13. Liu Y, Zheng P, Wu H, Zhang Y. Preparation and dynamic moisture adsorption of fiber felt/silica aerogel composites with ultra-low moisture adsorption rate. *Construction and Building Materials* **363**, 129825 (2023).
 14. Jia C, *et al.* Highly compressible and anisotropic lamellar ceramic sponges with superior thermal insulation and acoustic absorption performances. *Nat Commun* **11**, 3732 (2020).
 15. Su L, *et al.* Anisotropic and hierarchical SiC@SiO₂ nanowire aerogel with exceptional stiffness and stability for thermal superinsulation. *Sci Adv* **6**, eaay6689 (2020).
 16. Grzybowski BA, Sobolev YI, Cybulski O, Mikulak-Klucznik B. Materials, assemblies and reaction systems under rotation. *Nat Rev Mater* **7**, 338-354 (2022).
 17. Evans G, Greif RJNHT, Part A: Applications. Effects of boundary conditions

- on the flow and heat transfer in a rotating disk chemical vapor deposition reactor. **12**, 243-252 (1987).
18. Hu B, *et al.* Experimental investigation on the flow and flow-rotor heat transfer in a rotor-stator spinning disk reactor. **162**, 114316 (2019).
 19. Pask SD, Nuyken O, Cai ZJPC. The spinning disk reactor: an example of a process intensification technology for polymers and particles. **3**, 2698-2707 (2012).
 20. Yoon MS, Hyun JM, Park JSJjoh, flow f. Flow and heat transfer over a rotating disk with surface roughness. **28**, 262-267 (2007).
 21. He YL, Xie T. Advances of thermal conductivity models of nanoscale silica aerogel insulation material. *Appl Therm Eng* **81**, 28-50 (2015).
 22. Yang H, *et al.* Reconstruction of Inherent Graphene Oxide Liquid Crystals for Large-Scale Fabrication of Structure-Intact Graphene Aerogel Bulk toward Practical Applications. *ACS Nano* **12**, 11407-11416 (2018).
 23. Zong D, *et al.* Bubble templated flexible ceramic nanofiber aerogels with cascaded resonant cavities for high-temperature noise absorption. *ACS Nano* **16**, 13740-13749 (2022).
 24. Pang K, *et al.* Highly efficient cellular acoustic absorber of graphene ultrathin drums. *Adv Mater* **34**, 2103740 (2022).
 25. Qiu L, Liu JZ, Chang SLY, Wu Y, Li D. Biomimetic superelastic graphene-based cellular monoliths. *Nature Communications* **3**, 1241 (2012).
 26. Yu ZL, *et al.* Superelastic hard carbon nanofiber aerogels. *Adv Mater* **31**, e1900651 (2019).
 27. Bryning MB, Milkie DE, Islam MF, Hough LA, Kikkawa JM, Yodh AG. Carbon Nanotube Aerogels. *Adv Mater* **19**, 661-664 (2007).
 28. Zong D, *et al.* Flexible ceramic nanofibrous sponges with hierarchically entangled graphene networks enable noise absorption. *Nat Commun* **12**, 6599 (2021).
 29. Dou L, *et al.* Interweaved cellular structured ceramic nanofibrous aerogels with superior bendability and compressibility. *Adv Func Mater*, 2005928

(2020).

30. Wang F, Dai J, Huang L, Si Y, Yu J, Ding B. Biomimetic and Superelastic Silica Nanofibrous Aerogels with Rechargeable Bactericidal Function for Antifouling Water Disinfection. *ACS Nano*, (2020).
31. Si Y, *et al.* Superelastic and superhydrophobic nanofiber-assembled cellular aerogels for effective separation of oil/water emulsions. *ACS Nano* **9**, 3791-3799 (2015).
32. Dong X, Si Y, Chen C, Ding B, Deng H. Reed Leaves Inspired Silica Nanofibrous Aerogels with Parallel-Arranged Vessels for Salt-Resistant Solar Desalination. *ACS Nano* **15**, 12256-12266 (2021).
33. Dou L, *et al.* Hierarchical Cellular Structured Ceramic Nanofibrous Aerogels with Temperature-Invariant Superelasticity for Thermal Insulation. *ACS Applied Materials & Interfaces* **11**, 29056-29064 (2019).
34. Dou L, *et al.* Hierarchical cellular structured ceramic nanofibrous aerogels with temperature-invariant superelasticity for thermal insulation. *ACS Appl Mater Interfaces* **11**, 29056–29064 (2019).

Reviewers' comments:

Reviewer #1 (Remarks to the Author):

The revised manuscript has addressed most of our comments and the quality has been improved significantly. However, there remain a few comments/suggestions as follows.

1. On Comment 3: Although the authors provided the total freezing times for different freezing methods, it is still a concern of making direct comparison between different approaches although these samples have distinct microstructures.
2. In view of the capability for large-scale production of aerogels, the most critical limitation of conventional freeze-casting technique is the maximum freezing distance with uniform properties. As such, the freezing distance should be considered the parameter for comparison rather than weight of ice block (Fig R1e). In addition, there is not much time difference in ice block weight less than 800g between the directional freezing and crushed ice casting. Explain why.
3. Fig. 4g: The thermal conductivity of lamellar sponges in the axial direction (circle) is found lower than that in the transverse direction (triangle) at all temperatures studied. Explain why and provide more convincing discussion.
In addition, anisotropic aerogels have been proven to have better insulation performance than their isotropic counterparts due to the improved thermal management by dissipating the heat in the axial direction and inhibiting the heat conduction in the transverse direction [J. Chem. Eng., 2020, 385, 123963; ACS Sustain. Chem. Eng., 2021, 9, 9348; Compos. B. Eng., 2022, 243, 110161]. However, a completely opposite observation is made in this work. The authors should provide a more convincing discussion.

Minor comments:

1. There are many typographical mistakes in the revision. For example, Fig. 4g, the figure legend: "anisotropic aerogels (axial)" and "anisotropic aerogels (transverse)". "Supplementary Table 1. Whole process time from raw materials to products comprsion between istropic ASNF aerogels..."
Supplementary Table 1, left column title: "Anisotropic ASNF..."
2. Caption for supplementary Fig. 7 c-h: the density of "0.59 mg/cm³" for (c) is missing.

Reviewer #2 (Remarks to the Author):

I went through the revised version of the manuscript. Although the authors replies to the points raised, I found many of these arguments still unconvincing and/or based on unfair comparisons. Regarding the freezing process itself, for example, which is supposedly the central novelty of the paper, the key argument proposed by the author is that the overall freezing time is faster, and this should enable commercialization of these materials. I think the authors do not have sufficient understanding of the costs and hurdles of commercializing novel materials. Freezing is (energy-wise) relatively cheap, the expensive part is the drying, which is the same in both cases (previous work and current work), and the cost of the process is just a small part of the equation to decide whether a novel material can be commercialized succesfully or not. Besides, the authors already mentioned in the introduction that freeze-casting is low cost.

The main benefit of the novel approach (claimed by the authors) is that pieces of large dimensions can be processed, but the same dimensions can be obtained with standard freeze-casting. The scientific advances are thus incremental at best, and there are still unfortunately still little exploration of the science behind the proposed approach. The proposed explanations (decoupling of the nucleation and growth phases) are not very convincing either, as the control of nucleation in « standard » freeze-casting have been achieved with many different strategies (including epitaxial growth of ice from a pre-existing ice template).

The key idea upon which the paper is based (preparing ice crystals and then mixing them with the solution) has already been reported in many papers: see for instance <https://doi.org/10.1002/mabi.200900468>, <https://doi.org/10.1177/0883911513494620>, or <https://doi.org/10.1002/adma.201200237>.

Alternatively, freeze-casting methods to process isotropic materials have already been reported as well (e.g. <https://doi.org/10.1016/j.msec.2012.08.004>)

The only novelty (in terms of process) that I see here is thus that it's made at a larger scale than before, which again is nice but very incremental in my opinion.

Overall, although the claimed novelty is on the process (bearing in mind my concerns above), it still represents only 20% of the paper or so, the rest is dedicated to different materials obtained by the process, I thus still find the paper poorly balanced.

Other comments:

- non-scientific expressions are still present throughout the manuscript (e.g. « salad-like », which does not mean anything, « ultralight feathers » ?)
- my concerns about reproducibility have not really been addressed, I still don't see error bars in Fig S7a, for example.
- non-standard tests (Fig 3a) are still present. These are uninformative since we cannot compare the behavior with that of previously reported materials.
- Fig 4g and 4i proves that the thermal conductivity properties are essentially the same than previously reported, which is not surprising since macropores dominate the thermal behaviors in both cases. There are thus no improvements here.
- the discussion section is actually the conclusion
- the comparison of the freezing time (Fig 2b and 2g) are misleading, since the starting temperatures are different in both cases ! If the standard freeze-casting was performed by starting with a suspension at 0°C, the total freezing time would be much shorter.
- typo in Fig 3f ("stroage")

Responses to referees

Dear referees,

We appreciate the constructive comments on our manuscript entitled “*Large-scale assembly of isotropic nanofiber aerogels based on columnar-equiaxed crystal transition* (Tracking #: *NCOMMS-22-33561C*)”. We are grateful for the time and effort the reviewers dedicated to providing feedback on our manuscript. The constructive comments and suggestions are very helpful for us to improve the manuscript.

According to these comments and suggestions from two referees, we have carefully revised the manuscript with all the changes highlighted. The comments are reproduced and our responses are given directly in a different color (blue). The point-by-point responses to Reviewer #1 and Reviewer #2 are listed in the following pages. All page numbers refer to the revised manuscript file with tracked changes.

Responses to Referee #1

Comment 1: The revised manuscript has addressed most of our comments and the quality has been improved significantly. However, there remain a few comments/suggestions as follows. Although the authors provided the total freezing times for different freezing methods, it is still a concern of making direct comparison between different approaches although these samples have distinct microstructures. In view of the capability for large-scale production of aerogels, the most critical limitation of conventional freeze-casting technique is the maximum freezing distance with uniform properties. As such, the freezing distance should be considered the parameter for comparison rather than weight of ice block (Fig R1e). In addition, there is not much time difference in ice block weight less than 800g between the directional freezing and crushed ice casting. Explain why?

Response: We are thankful for the constructive comments. As the reviewer suggested, we have deleted the comparison of the freezing time between two different approaches. We are more concerned about the time required to form ice blocks with the same thickness. We have provided a comparison of the freezing distance to replace the weight of the ice block to prove the capability for large-scale production of aerogels (Supplementary Fig. 10).

When comparing time consumption between different methods, the set freezing area and final freezing environment are constant. When the ice block weights less than 800g, the thickness of the liquid film is 2 cm. In this case, the heat transfer resistance is relatively high. The freezing rate gradually decreases from bottom to top due to the gradual vanishing of temperature gradient and reduction of heat transfer. This thickness is much greater than the liquid film thickness on the surface of the rotating cryogenic drum (~3 mm). Although decoupling the ice nucleation and growth processes can enhance the freezing process, the crushed ice casting will not exhibit particularly significant advantages at lower thicknesses owing to the two-step program. With the increase in freezing distance, the upper solution is far away from the cold plate. And the freezing rate of the upper solution significantly decreases, resulting in a slow nucleation rate. Therefore, the freezing time increases rapidly with the increase in thickness.

Supplementary Fig. 10. Comparison of time consumption between crushed ice casting and unidirectional freezing methods. a Total production time comparison. **b** Time consumption of crushed ice manufacturing (first step) and crushed ice re-casting steps (second step). **c** Freezing rate of crushed ice re-casting and unidirectional freezing.

Revise details:

- Page 9, Line 3, added, “Next, we compared the overall freezing efficiency of different freezing methods. When the freezing distance was low, the CIC method had no significant advantage over directional freezing because two separate steps were required. However, when producing ice block with a high thickness of 3 cm, the total freezing time was 1.38 times shorter than that of the conventional unidirectional freezing process, and 1.92 times shorter than that of the nondirectional freezing process, (Supplementary Fig. 10). With an increase of freezing distance, the freezing efficiency would be more significant. This can be explained as the process intensification from decoupling the ice nucleation and growth processes, including: (i) Thin film freezing process on rotating cryogenic drum can effectively increase heat transfer efficiency; and (ii) Multi-point growth from 3D distribution of crystal nucleus can reducing crystallization path and accelerating the crystallization efficiency.”

Comment 2: Fig. 4g: The thermal conductivity of lamellar sponges in the axial direction (circle) is found lower than that in the transverse direction (triangle) at all temperatures studied. Explain why and provide more convincing discussion. In addition, anisotropic aerogels have been proven to have better insulation performance than their isotropic counterparts due to the improved thermal management by dissipating the heat in the

axial direction and inhibiting the heat conduction in the transverse direction [J. Chem. Eng., 2020, 385, 123963; ACS Sustain. Chem. Eng., 2021, 9, 9348; Compos. B. Eng., 2022, 243, 110161]. However, a completely opposite observation is made in this work. The authors should provide a more convincing discussion.

Response: Thank you for your meaningful comments and suggestions. I am very sorry that we may not clearly describe the structure of the lamellar sponge (*Nat Commun* 11, 3732 (2020); *ACS Nano* 12, 3103-3111 (2018).), which is stacked layer by layer. The axial and radial direction is completely different from the anisotropic aerogel (Supplementary Fig. 15). The thermal conductivity of the lamellar sponges consists of three parts: thermal convection, thermal conduction and thermal radiation. The low thermal conductivity of the axial direction is mainly provided by submicro-fibrous building block, lamellar structure, and ultra-light characteristics, owing to the layer-by-layer air blocking effect, the multilayer diffuse reflection effect, and the thermal bridge inhibition effect¹. However, in the radial direction, gas conduction and convection mainly contribute to the thermal conductivity in lamellar sponges owing to the large gaps between lamellas, while the nanofiber walls serve as channels for solid conduction.

We agree with the reviewer's viewpoint of anisotropic aerogels have better insulation performance than their isotropic counterparts, which is not contradictory to our results. The thermal conductivity of the anisotropic aerogels in the radial direction is relatively lower than that of the axial direction, which is may also smaller than the porous structure formed by random freezing. When the thermal energy is propagated in the radial direction, lamellar aerogels prepared by unidirectional freezing could effectively block solid heat conduction since the interlamellar connection is missing. Moreover, the thermal convection is also restricted in the interlaminar, while the thermal radiation could be weakened by multiple refraction and emission between the regular lamellas, thus significantly reducing the total thermal conductivity². In the axial direction, heat would transfer through the gap between lamellas, resulting in higher thermal conductivity as compared with that in radial direction. As stated by the referee, this phenomenon has been confirmed by many literatures. Furthermore, as compared with isotropic aerogel (Random), anisotropic thermal properties of aerogel can enable efficient thermal dissipation along the axial direction, thus yielding improved thermal insulation in the radial direction. A very complete and accurate explanation has been provided by the literatures (Chem. Eng. J., 2020, 385, ACS Sustain. Chem. Eng., 2021,

9, 9348; Compos. B. Eng., 2022, 243, 110161),

It must be pointed out that, the manifestation of advantages must be reflected in the basically identical pore structure, including size and topology. In the literature description (Chem. Eng. J., 2020, 385), their random freezing is not crushed ice casting, ice crystals still grow along temperature gradients (from the outside to the inside), forming a seemingly isotropic structure. Columnar crystals dominate this structure, making it difficult to form a more refined separate cellular structure.

In terms of CIC process, ice crystals are artificially added to increase discrete nuclei. All pre-existing and 3D random distributed ice crystals can act as primitive growth sites. The adjacent crystals will unconstrainedly grow with no favorable location or orientations and ultimately form multi-domain bulk materials, along with the refinement of individual ice crystals. After the sublimation of ice crystals, the structure distributed along the ice crystal interface assembled by nanofibers with high tortuous is preserved. An isotropic aerogel with separate cellular networks can be readily obtained. The refined tortuous and complex pore structure would become effective barrier for thermal transport. Thus, the thermal conductivity of our structure is lower than that of anisotropic aerogel.

For anisotropic aerogels, heat will flow through parallel channels in the axial direction and be transferred to the outside of the material for heat dissipation. However, due to insufficient lateral dimensions, anisotropic materials are difficult to utilize in the radial direction for practical insulation applications. While for ASNFs with unrestricted available size, this separate cellular pore structure can effectively block heat in the cell cavity and prevent it from conducting. As the reviewer noted, we have provided more convincing discussion about the thermal conductivity of lamellar sponges in different direction.

Supplementary Fig. 15. Structure comparison of different porous materials. a SEM images of the ASNF lamellar sponges from the different view. **b** 3D reconstruction of the ASNF anisotropic aerogels from X-ray microtomography (upper). SEM images from the top view at different magnifications (bottom). **c** 3D structure of the ASNF isotropic aerogels from X-ray microtomography.

Revise details:

- Page 12, Line 10, added “The low thermal conductivity of the axial direction was mainly provided by the the thermal bridge inhibition effect, layer-by-layer air blocking effect, and the multilayer diffuse reflection effect¹. However, in the radial direction, gas conduction and convection mainly contributed to the thermal conductivity owing to the large gaps between lamellas, while the nanofiber walls serve as channels for solid conduction.”

- Page 12, Line 19, added “Anisotropic thermal properties of aerogel can enable efficient thermal dissipation along the axial direction, thus yielding enhanced thermal insulation in the radial direction.”
- Page 12, Line 25, added “The reason can be explained as the columnar-equiaxed crystal transition in the freezing procedure resulting in thousands of repeating refined microscale units with a tortuous channel topology, reducing gaseous thermal conductivity and length of solid heat conduction path.”

Comment 3: There are many typographical mistakes in the revision. For example,

Fig. 4g, the figure legend: “anisotropic aerogels (axial)” and “anisotropic aerogels (transverse)”.

“Supplementary Table 1. Whole process time from raw materials to products comparison between isotropic ASNF aerogels...”

Supplementary Table 1, left column title: “Anisotropic ASNF...”

Response: Thank you for the comments. We are sorry for the careless mistakes. We have revised them.

Revise details:

- Fig. 4g, the figure legend “anisotropy” was revised as “anisotropic”
- Supplementary Table 2. “isotropic ASNF aerogels...” was revised as “isotropic ASNF aerogels...”
- Supplementary Table 2, left column title: “Astropic ASNF aerogels” was revised as “Isotropic ASNF aerogels”

Responses to Referee #2

Comment 1: I went through the revised version of the manuscript. Although the authors reply to the points raised, I found many of these arguments still unconvincing and/or based on unfair comparisons. Regarding the freezing process itself, for example, which is supposedly the central novelty of the paper, the key argument proposed by the author is that the overall freezing time is faster, and this should enable commercialization of these materials. I think the authors do not have sufficient understanding of the costs and hurdles of commercializing novel materials. Freezing is (energy-wise) relatively cheap, the expensive part is the drying, which is the same in both cases (previous work and current work), and the cost of the process is just a small part of the equation to decide whether a novel material can be commercialized successfully or not. Besides, the authors already mentioned in the introduction that freeze-casting is low cost. The main benefit of the novel approach (claimed by the authors) is that pieces of large dimensions can be processed, but the same dimensions can be obtained with standard freeze-casting.

Response: We are thankful for these comments. We are sorry for our responses did not meet the point. We agree with the referee's viewpoint on the energy consumption of freezing is a small part of the freeze-drying process (~5%). Solving the freezing problem cannot solve the industrialization problem of our materials. However, it is worth mentioning that our work does own several innovative points more than the freezing time. We would like to emphasize our innovation to present this work's novelty and research focus.

(1) **Decoupling the ice nucleation and growth processes in freezing process.** We successfully combine two mechanisms to enhanced the freezing process through decoupling the ice nucleation and growth processes. (i) Thin film freezing process on rotating cryogenic drum can effectively increase heat transfer efficiency; and (ii) Multi-point growth from 3D distribution of crystal nucleus can reducing crystallization path and accelerating the crystallization efficiency. Freezing is a universal process, this method may inspire other freezing and even crystallization processes to increase efficiency and reduce energy consumption. (Also mentioned by Referee 1)

(2) **Columnar-equiaxed crystal transition.** In the manufacturing process of industrial ingots, columnar crystals seriously affect the strength and toughness of materials.

To increase the proportion of equiaxed grains, mechanical vibration or ultrasound inducing were applied to form more discrete nuclei. Similar in the crushed ice casting process, ice crystals are artificially added to increase discrete nuclei, thereby promoting the formation of similar equiaxed crystals. In our process, the adjacent crystals will unconstrainedly grow with no favorable location or orientations and ultimately form multi-domain bulk materials, along with the refinement of individual ice crystals. After the sublimation of ice crystals, the structure distributed along the ice crystal interface assembled by nanofibers with high tortuous is preserved. An isotropic aerogel with separate cellular networks can be readily obtained. The refined tortuous and complex pore structure would become effective barrier for thermal, electronic, or acoustic transport.

- (3) ***New material space.*** The crushed ice casting method can also construct lightweight structures from various low-dimensional materials. Owing to the rapid preparation and large proportion introduction of crushed ice, this method is less prone to particle sedimentation in preparing large-sized samples, exhibiting high compatibility with different low-dimensional materials compared to traditional methods. We successfully prepared 3D porous aerogels from the snowflakes containing aluminosilicate nanofibers (ASNFs), polyacrylonitrile nanofibers (PNFs), p-aramid nanofibers (ANFs), cellulose nanofibers (CNFs), and graphite nanotubes (GNTs)). (Also mentioned by Editor).
- (4) ***Important application.*** We performed thermal propagation experiments to show important application of the aerogels as reliable thermal protector under fire and explosion. The complex thermo-mechanical shock in battery fires with a $\text{LiNi}_{0.8}\text{Co}_{0.1}\text{Mn}_{0.1}\text{O}_2$ (NCM811) cathode has been considered as the most complicated hazards among other battery TR events. Taking advantages of its stable mechanical and thermal insulation performance at ultrahigh temperatures, a thin layer of the aerogels can successfully prevent the deflagration propagation of a pack of high-energy Li-ion batteries, while one of the batteries caught on fire/explosion. As far as we know, no materials have been able to achieve this effect. Our materials provide new technique to solve the energy storage safety problems, which is the primary focus of the electric vehicle (EV) community. (Also mentioned by Referee 3).

According to your comments, we have made the following revise:

- Page 7, Line 30, we added, “However in CIC process, the setting temperature and slurry temperature were essentially consistent and no nucleation peak appeared, demonstrating the crystallization process does not require excessive external energy input to overcome the crystallization energy barrier (Fig. 2b).
- Page 9, Line 16, we added, “In terms of resultant materials, the columnar and equiaxed crystals were main components in ice blocks from unidirectional freezing and CIC, respectively derived from the difference in the growth orientation. In the manufacturing process of industrial ingots, columnar crystals seriously affect the strength and toughness of materias. To increase the proportion of equiaxed grains, mechanical vibration or ultrasound inducting were applied to form more discrete nuclei. Similar in the CIC process, ice crystals were artificially added to increase discrete nuclei, thereby promoting the formation of similar equiaxed crystals. We investigated the influence of ice-slurry ratio on the crystal structure of obtained ice blocks. When the proportion of ice crystals was small, most ice crystals would form oriented columnar crystals along the direction of temperature gradient. Only a small portion of the area was affected by external ice crystals to form partial equiaxed crystals (Fig. 2d). When the proportion of ice crystals reached a reasonable level (2 : 1), all pre-existing and 3D randomly distributed ice crystals could act as primitive growth sites. The adjacent crystals would unconstrainedly grow with no favorable location or orientations and ultimately form multi-domain bulk materials, along with the refinement of individual ice crystals. After the sublimation of ice crystals, the structure distributed along the ice crystal interface assembled by nanofibers with high tortuous was preserved. An isotropic aerogel with separate cellular networks could be readily obtained. The refined tortuous and complex pore structure would become effective barrier for thermal, electronic, or acoustic transport.”
- Page 5, Line 21, we added, “Owing to the rapid preparation and large proportion introduction of crushed ice, this method is less prone to particle sedimentation in

preparing large-sized samples, exhibiting high compatibility with different low-dimensional materials compared to traditional methods.”

Comment 2: The scientific advances are thus incremental at best, and there are still unfortunately still little exploration of the science behind the proposed approach. The proposed explanations (decoupling of the nucleation and growth phases) are not very convincing either, as the control of nucleation in « standard » freeze-casting have been achieved with many different strategies (including epitaxial growth of ice from a pre-existing ice template).

Response: Thank you for your meaningful comments. Indeed, we need to make the scientific advances clearer.

The decoupling mechanism of nucleation and growth proposed by us is supported by theoretical foundations. It is mainly based on the solidification mechanism of equiaxed crystal in ingots, and the nucleation theory has a long history of development and is therefore relatively mature. The phenomenon of columnar crystal to equiaxed crystal transition (CET) occurs in the solidification process of actual production. In 1966, Jackson K.A. observed the phenomenon of dendritic melting in the alloy, and the resulting free crystals were acted as sources from equiaxed grains during the CET process. Afterwards, Southin proposed the "Crystalline rain" theory. And Tokumi Ono believed that the detachment and proliferation of ice crystal after detachment from mould wall are the sources of equiaxed nuclei. At present, these three theories are also the main part of equiaxed crystal zone forming. The common feature is the generation of excess crystal nuclei, the source of crystal nuclei for subsequent equiaxed crystal, in the central part under different actions(10.1179/cmq.1969.8.2.189, 10.2355/isijiinternational.43.1415).

Based on this theory, increasing the content of fractured dendrites under external forces to increase nucleation points and refine grains is also a commonly used method in the ingot casting process. Under the action of vibration, convection intensifies during the solidification process of metal, causing relative motion between the liquid and solid phases, thereby promoting the fracture or detachment of dendrites at the liquid-solid interface from the matrix and becoming the source of fine equiaxed crystal nuclei. (<https://doi.org/10.1080/03019233.2017.1364904>).

Therefore, we propose a coupling mechanism between nucleation and crystal growth based on the above theory. The added ice particles can serve as a source of nucleation for the subsequent freezing process, thereby saving nucleation time. At the same time,

the introduction of ice particles increases the nucleation sites during the freezing process, thereby regulating the growth orientation of ice crystals and forming similar randomly distributed equiaxed ice crystals. Finally, a randomly distributed pore structure was obtained through freeze-drying. Based on the suggestions of the reviewers, we have rewritten the relevant paragraphs and added appropriate descriptions of nucleation. We have also attempted to clarify the theoretical basis for our proposed mechanism more clearly.

Fig. 2. Proposed freezing mechanism. **a** Schematic illustration of the unidirectional freezing (upper) and crushed ice casting (bottom) processes. **b** Temperature variation over time during unidirectional freezing (upper) and crushed ice casting (bottom) . **c** *In situ* observation on unidirectional freezing (upper) and crushed ice casting (bottom) using a optical-fluorescence microscope. **d** *In situ* observation on crushed ice casting using a optical-fluorescence microscope in different mass ratio for ice crystal to slurry.

Revise details:

- In Page 7, Line 24, we added, “To probe the mechanism differences between unidirectional freezing and CIC, we *in-situ* observed the cooling process under an optical-fluorescence microscope after mixing with a small amount of fluorescent polystyrene microspheres. Two comparative tests have been conducted at a same cooling rate of $1^{\circ}\text{C min}^{-1}$ during freezing. In unidirectional freezing process, the slurry temperature decreased linearly before ice nucleation. A sudden temperature rise could be observed when the slurry temperature reached 1.3°C , which was attributed to the heat release upon nucleation. However in CIC process, the setting temperature and slurry temperature were essentially consistent and no nucleation peak appeared, demonstrating the crystallization process does not require excessive external energy input to overcome the crystallization energy barrier (Fig. 2b).”
- In Page 9, Line 3, we added, “Next, we compared the overall freezing efficiency of different freezing methods. When the freezing distance was low, the CIC method had no significant advantage over directional freezing because two separate steps were required. However, when producing ice block with a high thickness of 3 cm, the total freezing time was 1.38 times shorter than that of the conventional unidirectional freezing process (Supplementary Fig. 10). With an increase of freezing distance, the freezing efficiency will be more significant. This can be explained as the process intensification from decoupling the ice nucleation and growth processes, including: (i) Thin film freezing process on rotating cryogenic drum can effectively increase heat transfer efficiency; and (ii) Multi-point growth from 3D distribution of crystal nucleus can reducing crystallization path and accelerating the crystallization efficiency. Moreover, the whole process time from raw materials to products was also much lower than the traditional fiber felt/silica aerogel composites (Supplementary Table 2), demonstrating the practicality of this method.”
- In Page 9, Line 16, we added, “In terms of resultant materials, the columnar and equiaxed crystals were main components in ice blocks from unidirectional freezing and CIC, respectively, derived from the difference in the growth orientation. In the manufacturing process of industrial ingots, columnar crystals

seriously affect the strength and toughness of materials. To increase the proportion of equiaxed grains, mechanical vibration or ultrasound inducing were applied to form more discrete nuclei. In the CIC process, ice crystals were artificially added to increase discrete nuclei, thereby promoting the formation of similar equiaxed crystals. We investigated the influence of ice-slurry ratio on the crystal structure of obtained ice blocks. When the proportion of ice crystals was small, most ice crystals would form oriented columnar crystals along the direction of temperature gradient. Only a small portion of the area was affected by external ice crystals to form partial equiaxed crystals (Fig. 2d). When the proportion of ice crystals reached a reasonable level (2 : 1), all pre-existing and 3D randomly distributed ice crystals could act as primitive growth sites. The adjacent crystals would unconstrainedly grow with no favorable location or orientations and ultimately form multi-domain bulk materials, along with the refinement of individual ice crystals. After the sublimation of ice crystals, the structure distributed along the ice crystal interface assembled by nanofibers was preserved. An isotropic aerogel with separate cellular networks could be readily obtained. The refined tortuous and complex pore structure would become effective barrier for thermal, electronic, or acoustic transport.”

Comment 3: The key idea upon which the paper is based (preparing ice crystals and then mixing them with the solution) has already been reported in many papers: see for instance <https://doi.org/10.1002/mabi.200900468>, <https://doi.org/10.1177/0883911513494620>, or <https://doi.org/10.1002/adma.201200237>. Alternatively, freeze-casting methods to process isotropic materials have already been reported as well (e.g. <https://doi.org/10.1016/j.msec.2012.08.004>) The only novelty (in terms of process) that I see here is thus that it's made at a larger scale than before, which again is nice but very incremental in my opinion.

Response: Thank the reviewer for the critical comments. We agree with the reviewer that the fabrication of porous materials by freeze-drying after mixing the ice particulates was not new. The obtained uniform porous materials have been proven to be applicable to tissue scaffolds. These methods partly overcoming the issues of low nucleation and growth efficiency in freeze casting process. However, such behavior, combined with process intensification and structural evolution, has not been verified nor extended to different material systems. Most critically of all, the fundamental question of how ice crystal transformation can allow such a large alteration in structure remains unanswered. The detailed demonstration summary of its novelty and characteristics is as follows.

- (1) In terms of production methods, these materials prepare initial ice templates by freezing pre-formed water droplets, and accurately control the shape and position of the initial ice crystals to regulate the pore structure of the final material. While the CIC method for producing ice crystals proposed in our work is more convenient, which can quickly and efficiently prepare crushed ice. Thereby a large amount of crushed ice can obtain in a short period of time.
- (2) There are also differences in the composition of ice crystals. During the preparation process, the dispersed solution is directly prepared into crushed ice instead of deionized water, so the density of the final block will not change significantly with the proportion of ice added.
- (3) There are differences in the mixing process between ice crystals and water. Due to the composition of ice crystals, there is no need to pre-freeze the mixed solution or add other reagents to avoid the melting of ice crystals during the mixing process. We significantly simplified the overall preparation process by adjusting the ratio of

ice to water.

- (4) The application scopes are varied. The work reported mostly is to prepare specific materials, which has precise requirements for structure. As for cell culture, it is required to have a uniform and regular distribution of pores to ensure the uniform distribution of subsequent cells. The method we proposed is a universal process for preparing isotropic porous materials, which can be applied to various fiber or powder materials that can be prepared into slurries.

As the reviewer suggested, we have cited relevant literature. And more detailed description was provided on the differences from previous work and the novelty were added in the revised manuscript.

Revise details:

- In Page 3, Line 17, we added “Alternatively, dynamic freeze casting and ice particulate templating can achieve multi-point ice nucleation and growth³⁻⁵. The obtained uniform porous materials have been proven to be applicable to tissue scaffolds. These methods partly overcoming the issues of low nucleation and growth efficiency in freeze casting process. However, such behavior, combined with process intensification and structural evolution, has not been verified nor extended to different material systems. Most critically of all, the fundamental question of how ice crystal transformation can allow for such a large alteration in structure remains unanswered.”

Comment 4: Overall, although the claimed novelty is on the process (bearing in mind my concerns above), it still represents only 20% of the paper or so, the rest is dedicated to different materials obtained by the process, I thus still find the paper poorly balanced.

Response: Thank you for your constructive comments and suggestions. We have deleted part of discussion on the characterization of material properties and placed the figures in supporting information, while adding a detailed description of the preparation process. More than 60% of the manuscript now revolves around preparation methods rather than material properties. As mentioned in Comment 1, we have conducted in-depth discussions on the novelty of the method.

Revise details:

- We deleted the part of “Temperature-invariant mechanical performances”.
- In Page 7, Line 24, we added, “To probe the mechanism differences between unidirectional freezing and CIC, we *in-situ* observed the cooling process under an optical-fluorescence microscope after mixing with a small amount of fluorescent polystyrene microspheres. Two comparative tests have been conducted at a same cooling rate of $1^{\circ}\text{C min}^{-1}$ during freezing. In the unidirectional freezing process, the slurry temperature decreased linearly before ice nucleation. A sudden temperature rise could be observed when the slurry temperature reached 1.3°C , which was attributed to the heat release upon nucleation. However, in CIC process, the setting temperature and slurry temperature were essentially consistent and no nucleation peak appeared, demonstrating the crystallization process does not require excessive external energy input to overcome the crystallization energy barrier (Fig. 2b).”
- In Page 9, Line 3, we added, “Next, we compared the overall freezing efficiency of different freezing methods. When the freezing distance was low, the CIC method had no significant advantage over directional freezing because two separate steps were required. However, when producing ice block with a high thickness of 3 cm, the total freezing time was 1.38 times shorter than that of the conventional unidirectional freezing process (Supplementary Fig. 10). With an increase of freezing distance, the freezing efficiency would be more significant. This can be explained as the process intensification from decoupling the ice

nucleation and growth processes, including: (i) Thin film freezing process on rotating cryogenic drum can effectively increase heat transfer efficiency; and (ii) Multi-point growth from 3D distribution of crystal nucleus can reducing crystallization path and accelerating the crystallization efficiency. Moreover, the whole process time from raw materials to products was also much lower than the traditional fiber felt/silica aerogel composites (Supplementary Table 2), demonstrating the practicality of this method.”

- In Page 9, Line 16, we added, “In terms of resultant materials, the columnar and equiaxed crystals were main components in ice blocks from unidirectional freezing and CIC, respectively, derived from the difference in the growth orientation. In the manufacturing process of industrial ingots, columnar crystals seriously affect the strength and toughness of materias. To increase the proportion of equiaxed grains, mechanical vibration or ultrasound inducting were applied to form more discrete nuclei. Similar in the CIC process, ice crystals were artificially added to increase discrete nuclei, thereby promoting the formation of similar equiaxed crystals. We investigated the influence of ice-slurry ratio on the crystal structure of obtained ice blocks. When the proportion of ice crystals was small, most ice crystals would form oriented columnar crystals along the direction of temperature gradient. Only a small portion of the area was affected by external ice crystals to form partial equiaxed crystals (Fig. 2d). When the proportion of ice crystals reached a reasonable level (2 : 1), all pre-existing and 3D randomly distributed ice crystals could act as primitive growth sites. The adjacent crystals would unconstrainedly grow with no favorable location or orientations and ultimately form multi-domain bulk materials, along with the refinement of individual ice crystals. After the sublimation of ice crystals, the structure distributed along the ice crystal interface assembled by nanofibers with high tortuous was preserved. An isotropic aerogel with separate cellular networks could be readily obtained. The refined tortuous and complex pore structure would become effective barrier for thermal, electronic, or acoustic transport.”

Comment 5: non-scientific expressions are still present throughout the manuscript (e.g. « salad-like », which does not mean anything, « ultralight feathers » ?)

Response: Thank you for your valuable comments and suggestions. We have double-checked the manuscript and deleted all the non-scientific expressions (such as “salad-like”, “ultralight feathers”) in the revised manuscript to ensure that all the claims revised on properties and performance are scientific and objective.

Revise details:

- In Page 10, Line 27, We revised “feathers” as “features”
- In Page 18, Line 25, We deleted “salad-like.”
- In Page 18, Line 21, We deleted “salad-like.”

Comment 6: My concerns about reproducibility have not really been addressed, I still don't see error bars in Fig S7a, for example.

Response: Thank you for your constructive comments and suggestions. We are sorry for our responses did not address your concerns. We have added a detailed description of the preparation process and characterization of the sample to further validate repeatability.

First, the ratio of crushed ice to dispersion was discussed, and the rationality of choosing a 5:1 ratio was confirmed. Second, the selection range of the slurry and the final obtained sample density were discussed. For crushed ice was directly prepared from the slurry, the density of the sample is only related to the slurry concentration. The error bars in Fig S7a have been added in the revised manuscript. Third, the consistency of the structure and performance of large-sized samples was discussed, confirming the feasibility of their use in preparing large-sized samples.

Supplementary Fig. 5. Density regulation of ASNF aerogels. a Density of aerogels *versus* concentration of nanofiber dispersion. **b** Weight measurement of ASNF

aerogels. The sample with a volume of 18.0 cm^3 has a mass of 10.7 mg , which corresponds to a bulk density of 0.59 mg cm^{-3} . **c-h** Compressive stress-strain curves of ASNF aerogels with the densities of 1.0 , 2.5 , 5.0 , 12.0 , and 16.0 mg cm^{-3} .

Supplementary Fig. 10. Comparison of time consumption between crushed ice casting and unidirectional freezing methods. a Total production time comparison. **b** Time consumption of crushed ice manufacturing (first step) and crushed ice re-casting steps (second step). **c** Freezing rate of crushed ice re-casting and unidirectional freezing.

Comment 7: non-standard tests (Fig 3a) are still present. These are uninformative since we cannot compare the behavior with that of previously reported materials.

Response: Thank you for your valuable comments and suggestions. The purpose of the test is to verify the impact elasticity of the aerogel materials. The recovery speed of ASNF aerogels was measured by rebounding a falling steel ball (2.26 g), which was calculated as 764 mm/s^{-1} , demonstrating the rapid rebound ability of the aerogels. This method has been used in many carbon or organic materials (e.g., carbon nanofibrous aerogels (860 mm s^{-1})⁶, chitosan–graphene oxide monoliths (580 mm s^{-1})⁷, ceramic nanofibrous aerogels (860 mm s^{-1})⁸, mullite sponges ($1,233 \text{ mm s}^{-1}$)⁹). As the reviewer noted, it is not a standard test.

In order to test the impact resistance of materials, we used drop hammer impact testing and pendulum impact testing. However, due to the characteristics of nanofiber materials, the prepared aerogel shows flexibility, which is difficult to obtain data from the test. We also attempted Hopkinson rod testing, but the mechanical strength of this porous material is not sufficient to obtain experimental data. To make the paper more scientific, we have removed this test.

Fig. R1. (a) Steel ball weight and sample size used in rapid rebound test. (b) Device diagram of steel ball drop experiment.

Revise details:

- We removed Fig 3a and Corresponding videos.

Comment 8: Fig 4g and 4i proves that the thermal conductivity properties are essentially the same than previously reported, which is not surprising since macropores dominate the thermal behaviors in both cases. There are thus no improvements here.

Response: Thank you for your important comments and suggestions. At present, there are many kinds of thermal insulation materials, and aerogel is a common material. Because air is an ideal thermal insulator, the low thermal conductivities of aerogels mainly originate from the restricted heat transfer across the gas phase confined in the voids. The ASNFs aerogels prepared by crushed ice casting achieves low thermal conductivity through the intrinsic low thermal conductivity of ceramic materials and the structure with high porosity. And the high tortuosity channel structure is realized by columnar-equiaxed crystal transition, which further reduces the convective heat transfer of the gas phase.

- (1) Using other different testing methods, some previous work reported very low thermal conductivity. For example, an infrared camera was used to record the heat-conducting process of the materials after applying laser heat source¹⁰. Then the thermal conductivity was calculated. In addition, testing in an argon or vacuum environment can result in lower thermal conductivity, as the heat transfer performance of argon or vacuum is lower than that of air. In this paper, we use the standard test method in the air atmosphere. Under the standard test condition, the air thermal conductivity at room temperature is 25.52 mW/m K. The ASNF aerogels has reached an ideal level in thermal conductivity.
- (2) While maintaining low thermal conductivity, temperature resistance and mechanical properties are the areas worth breaking through in thermal insulation materials. We compared the lowest thermal conductivity and highest temperature resistance of our ASNF aerogels with those of other typical thermal insulation materials in an oxidizing atmosphere, such as nanocellulose/graphene oxide¹¹, nanowood¹², SiO₂ aerogels¹³, carbon nanotube aerogels¹⁴, graphene/Al₂O₃¹⁵, hBN aerogels¹⁶, SiC@SiO₂ nanowire aerogels¹⁰, SiO₂ nanofiber aerogels⁸, and ceramic microfiber sponges¹⁷ (Fig. 3i and Supplementary Table 5). The ASNF aerogels showed a lower thermal conductivity than most of the reported thermally insulating materials, well below that of standing air (0.025 W m⁻¹ K⁻¹). Furthermore, the long-term temperature resistance of ASNF aerogels was

1200 °C, which was the highest among the thermally superinsulating materials.

- (3) Most elastic structures with low thermal conductivity made from polymers or carbonaceous materials cannot withstand high temperatures under ambient conditions. As ceramic families, ASNF aerogels are expected to possess temperature-invariant elasticity. Different viscoelastic properties, such as storage modulus, loss modulus, and damping ratio, were investigated over a broad temperature range of -100 to 500 °C at a constant frequency of 1 Hz. Indeed, the aerogels showed temperature-independent stable viscoelastic performances and a consistent small damping ratio of ~ 0.1 (Supplementary Fig. 16a-c). A frequency dependency test (0.1 to 100 Hz) also showed stable viscoelastic properties over a wide temperatures range of -100 to 500 °C. Furthermore, compression tests were conducted by exposing the materials to a butane flame (1300 °C) and submerging it in liquid nitrogen (-196 °C) (Supplementary Fig. 16d and Video 8). The ASNF aerogels retained their resilience up to 80% compressive strain under extreme conditions. After several cycles, the aerogels fully recovered, with no obvious fracture.
- (4) We performed thermal propagation experiments to show important application of the aerogels as reliable thermal protector under fire and explosion. The complex thermo-mechanical shock in battery fires with a $\text{LiNi}_{0.8}\text{Co}_{0.1}\text{Mn}_{0.1}\text{O}_2$ (NCM811) cathode has been considered the most complicated hazards among other battery TR events. Taking advantages of its stable mechanical and thermal insulation performance at ultrahigh temperatures, a thin layer of the aerogels can successfully prevent the deflagration propagation of a pack of high-energy Li-ion batteries, while one of the batteries caught on fire/explosion. As far as we know, no materials have been able to achieve this effect. Our materials provide new technique to solve the energy storage safety problems, which is the primary focus of the electric vehicle (EV) community.

Comment 9: the discussion section is actually the conclusion

Response: Thank you very much for your important comments and suggestions. As suggested by the reviewer, we have increased the content proportion of discussion. And in terms of typesetting, we also strictly followed the requirements of the journal format and rewritten the discussion section (e. g. [10.1038/s41467-021-26890-9](https://doi.org/10.1038/s41467-021-26890-9); [10.1038/ncomms6802](https://doi.org/10.1038/ncomms6802))

Revise details:

- In Page 13, Line 18, From then on, we set up as Discussion.
- In Page 13, Line 32, we added “Most elastic structures with low thermal conductivity made from polymers or carbonaceous materials cannot withstand high temperatures under ambient conditions. As ceramic families, ASNF aerogels are expected to possess temperature-invariant elasticity. Different viscoelastic properties, such as storage modulus, loss modulus, and damping ratio, were investigated over a broad temperature range of -100 to 500 °C at a constant frequency of 1 Hz. Indeed, the aerogels showed temperature-independent stable viscoelastic performances and a consistent small damping ratio of ~ 0.1 (Supplementary Fig. 16a-c). A frequency dependency test (0.1 to 100 Hz) also showed stable viscoelastic properties over a wide temperatures range of -100 to 500 °C. Furthermore, compression tests were conducted by exposing the materials to a butane flame (1300 °C) and submerging it in liquid nitrogen (-196 °C) (Supplementary Fig. 16d and Video 8). The ASNF aerogels retained their resilience up to 80% compressive strain under extreme conditions. After several cycles, the aerogels fully recovered, with no obvious fracture. After a long-term high-temperature treatment (1200 °C for 24 h), the characteristic peak of mullite and cristobalite crystal phases appeared in the X-ray diffraction pattern (XRD) (Supplementary Fig. 17). With an average crystal size of 65.7 nm and crystallinity of 69% at this time, the nanograin-glassy dual-phase structure would not severely damage the strength and flexibility of ASNF aerogels¹. However, we could observe many defects formed on the nanofiber surface after calcination at 1300 °C; along with the decrease of elasticity and the increase of density, the nanofibers would gradually become brittle and finally be crushed. ”

Comment 10: The comparison of the freezing time (Fig 2b and 2g) are misleading, since the starting temperatures are different in both cases! If the standard freeze-casting was performed by starting with a suspension at 0°C, the total freezing time would be much shorter.

Response: Thank you for your valuable comments. We apologize for not expressing this well. We must emphasize that the comparison time we used is the total time of the two-step method rather than a specific step, as this comparison has been explained in Supplementary Fig. 10. The starting and ending standards for recording time are unified, both from room temperature slurry to block ice. In this section, we mainly focus on these points: firstly, the duration time of the crystallization process of the two freezing methods; secondly, the crystal morphology after crystallization; thirdly, whether it is necessary to overcome the crystal energy barrier.

Revise details:

- In Page 9, Line 2, we added “Next, we compared the overall freezing efficiency of different freezing methods. When the freezing distance was low, the CIC method has no significant advantage over directional freezing because two separate steps were required. However, when producing ice block with a high thickness of 3 cm, the total freezing time was 1.38 times shorter than that of the conventional unidirectional freezing process (Supplementary Fig. 10).”
- In Page 7, Line 24, we revised “To probe the mechanism differences between unidirectional freezing and CIC, we *in-situ* observed the cooling process under an optical-fluorescence microscope after mixing with a small amount of fluorescent polystyrene microspheres. Two comparative tests have been conducted at a same cooling rate of 1°C min⁻¹ during freezing. In unidirectional freezing process, the slurry temperature decreased linearly before ice nucleation. A sudden temperature rise could be observed when the slurry temperature reached 1.3 °C, which was attributed to the heat release upon nucleation. However in CIC process, the setting temperature and slurry temperature were essentially consistent and no nucleation peak appeared, demonstrating the crystallization process does not require excessive external energy input to overcome the crystallization energy barrier (Fig. 2b).”

Comment 11: typo in Fig 3f ("stroage")

Response: Thank you very much for your valuable comments and suggestions. We have revised it.

Revise details:

- Fig 3f, “stroage” was revised as “storage”

References

1. Li L, *et al.* Nanograin–glass dual-phasic, elasto-flexible, fatigue-tolerant, and heat-insulating ceramic sponges at large scales. *Mater Today* **54**, 72-82 (2022).
2. Zhang X, Zhao X, Xue T, Yang F, Fan W, Liu T. Bidirectional anisotropic polyimide/bacterial cellulose aerogels by freeze-drying for super-thermal insulation. *Chem Eng J* **385**, 123963 (2020).
3. Zhang Q, Lu H, Kawazoe N, Chen G. Preparation of collagen scaffolds with controlled pore structures and improved mechanical property for cartilage tissue engineering. *J Bioact Compatible Polym* **28**, 426-438 (2013).
4. Ko Y-G, Grice S, Kawazoe N, Tateishi T, Chen G. Preparation of collagen-glycosaminoglycan sponges with open surface porous structures using ice particulate template method. *Macromol Biosci* **10**, 860-871 (2010).
5. Jung H-D, Yook S-W, Jang T-S, Li Y, Kim H-E, Koh Y-H. Dynamic freeze casting for the production of porous titanium (Ti) scaffolds. *Materials Science and Engineering: C* **33**, 59-63 (2013).
6. Yu ZL, *et al.* Superelastic hard carbon nanofiber aerogels. *Adv Mater* **31**, e1900651 (2019).
7. Gao HL, *et al.* Super-elastic and fatigue resistant carbon material with lamellar multi-arch microstructure. *Nat Commun* **7**, 12920 (2016).
8. Si Y, Wang X, Dou L, Yu J, Ding B. Ultralight and fire-resistant ceramic nanofibrous aerogels with temperature-invariant superelasticity. *Sci Adv* **4**, eaas8925 (2018).
9. Kim KH, Oh Y, Islam MF. Graphene coating makes carbon nanotube aerogels superelastic and resistant to fatigue. *Nature Nanotechnology* **7**, 562-566 (2012).

10. Su L, *et al.* Anisotropic and hierarchical SiC@SiO₂ nanowire aerogel with exceptional stiffness and stability for thermal superinsulation. *Sci Adv* **6**, eaay6689 (2020).
11. Wicklein B, *et al.* Thermally insulating and fire-retardant lightweight anisotropic foams based on nanocellulose and graphene oxide. *Nat Nanotechnol* **10**, 277-283 (2015).
12. Li T, *et al.* Anisotropic, lightweight, strong, and super thermally insulating nanowood with naturally aligned nanocellulose. *Sci Adv* **4**, eaar3724 (2018).
13. Zhao S, *et al.* Additive manufacturing of silica aerogels. *Nature* **584**, 387-392 (2020).
14. Zhan HJ, *et al.* Biomimetic carbon tube aerogel enables super-elasticity and thermal insulation. *Chem* **5**, 1871-1882 (2019).
15. Zhang Q, *et al.* Flyweight, superelastic, electrically conductive, and flame-retardant 3D multi-nanolayer graphene/ceramic metamaterial. *Adv Mater* **29**, 1605506 (2017).
16. Xu X, *et al.* Double-negative-index ceramic aerogels for thermal superinsulation. *Science* **363**, 723-727 (2019).
17. Jia C, *et al.* Highly compressible and anisotropic lamellar ceramic sponges with superior thermal insulation and acoustic absorption performances. *Nat Commun* **11**, 3732 (2020).

REVIEWERS' COMMENTS

Reviewer #1 (Remarks to the Author):

The authors have addressed all my comments to my satisfaction, and thus the revision is now acceptable for publication in the Journal.

Responses to Referee #1

Comment 1: The authors have addressed all my comments to my satisfaction, and thus the revision is now acceptable for publication in the Journal.

Response: Thank you for the positive comments. We appreciate the constructive comments on our manuscript entitled “*Large-scale assembly of isotropic nanofiber aerogels based on columnar-equiaxed crystal transition* (Tracking #: NCOMMS-22-33561C)”. We are grateful for the time and effort the reviewers dedicated to providing feedback on our manuscript. The constructive comments and suggestions are very helpful for us to improve the manuscript.